# Modelling the response of Net Primary Productivity of Zambezi teak forests to climate change along a rainfall gradient in Zambia

Justine Ngoma[1,2], Maarten C. Braakhekke[4], Bart Kruijt[2], Eddy Moors[4,6], Iwan Supit[2], James H. Speer[3], Royd Vinya[1] & Rik Leemans[5]

[1]School of Natural Resources, The Copperbelt University P.O. Box 21692, Kitwe, Zambia.
[2]Water Systems and Global Change Group, Wageningen University and Research, P.O. Box 47, 6700AA Wageningen, The Netherlands
[3]Department of Earth and Environmental Systems, Indiana State University, Terre Haute, Indiana, 47809 USA
[4]VU University Amsterdam, De Boelelaan 1085, 1081 HV Amsterdam, The Netherlands
[5]Environmental Systems Analysis Group, Wageningen University and Research, P.O Box 47, 6700AA Wageningen, The Netherlands.
[6]IHE Delft Institute for Water Education, PO Box 3015, 2601 DA Delft, The Netherlands

*Correspondence to*: Justine Ngoma (justinangoma@yahoo.com)

**Abstract.** Understanding climate change effects on forests is important considering the role forests play in mitigating climate change. We studied the effects of changes in temperature, rainfall, atmospheric carbon dioxide ($CO_2$) concentration, solar radiation, and number of wet days (as a measure of rainfall intensity) on net primary productivity (NPP) of the Zambian Zambezi teak forests along a rainfall gradient. Using 1960-1989 as base-line, we projected changes in NPP for the end of the 21st century (2070-2099). We adapted the parameters of the dynamic vegetation model, LPJ-GUESS, to simulate the growth of Zambian forests at three sites along a moisture gradient receiving annual rainfall of between 700 mm to more than 1000 mm. The adjusted plant functional type was tested against measured data. We forced the model with contemporary climate data (1960-2005) and with climatic forecasts of an ensemble of five General Circulation Models (GCMs) following Representative Concentration Pathways (RCP) RCP4.5 and RCP8.5. We used local soil parameter values to characterize texture and measured local tree parameter values for maximum crown area, wood density, leaf longevity, and allometry. The results simulated with the LPJ-GUESS model improved when we used these newly generated local parameters indicating that using local parameter values is essential to obtaining reliable simulations at site level. The adapted model setup provided a baseline for assessing the potential effects of climate change on NPP in the studied Zambezi teak forests. Using this adapted model version, NPP was projected to increase by 1.77% and 0.69% at the wetter Kabompo, and by 0.44% and 0.10% at the intermediate Namwala sites under RCP8.5 and RCP4.5 respectively especially caused by the increased $CO_2$ concentration by the end of the 21st century. However, at the drier Sesheke site, NPP would respectively decrease by 0.01% and 0.04% by the end of the 21st century under RCP8.5 and RCP4.5. The projected decreased NPP under RCP8.5 at the Sesheke site results from the reduced rainfall coupled with increasing temperature. We thus demonstrated that differences in the amount of rainfall received in a site per year influence the way in which climate change will affect forests resources. The projected increase in $CO_2$ concentration would thus have more effects on NPP in high rainfall receiving areas, while in arid regions, NPP would be affected more by the changes in rainfall and temperature. $CO_2$ concentrations would therefore be more important in forests that are generally not temperature or precipitation limited, however precipitation will continue to be the limiting factor in the drier sites.

# 1    Introduction

The tropical Zambezi teak forests represent some of the most important forest types of southern Africa. They are distributed in Angola, Botswana, Namibia, Zambia, and Zimbabwe. These forests are a source of various ecosystem services including valuable commercial timber produced from *Baikiaea plurijuga* Harm (Piearce, 1986a; Piearce, 1986c). Additionally, the Zambezi teak forests play a substantial role in mitigating climate change as carbon sinks (Sarmiento and Gruber, 2002). This role is influenced by climate change through the mechanisms of forests' NPP. The effects of these climatic changes vary with location, ecosystem types, and climate zones (Wu et al., 2011). While increased temperature stimulates plant productivity to its optimal temperature in some plants (Wu et al., 2011) it also exponentially stimulates autotrophic plant respiration (Burton et al., 2008; Wu et al., 2011). Such increasing temperature effects can either be enhanced or moderated, depending on whether water availability decreases or increases (Chen et al., 2013). Reduced rainfall generally supresses the productivity of the plants (Wu et al., 2011).

In southern Africa, rainfall has declined (Hoerling et al., 2006; Niang et al., 2014) and dry spells have increased (New et al., 2006) over the last few decades. Model projections indicate that this trend will continue in the future. During the past half century, mean annual temperatures increased by 0.5 °C in some parts of Africa (Niang et al., 2014). By the end of the 21st century, southern African mean temperatures are projected to increase by between 3.4 ℃ and 4.2 ℃ above the 1981-2000 baseline under the A2 scenario (Niang et al., 2014).

In southern Zambia, maximum temperatures increased by 1 °C between 1976 and 2016 (Dube and Nhamo, 2018), and over the past 30 years, the Zambian mean temperatures increased by 0.6 °C (Bwalya, 2010). A 31 years of temperature records showed a substantial increase in average seasonal temperatures (October-April) (Mulenga et al., 2017). By the year 2070, Zambia's temperatures are projected to increase by 2.9 ℃ with reference to 1880 (The Government of the Republic of Zambia et al., 2007). Between 1976 and 2016, rainfall reduced by 47 mm in Southern Zambia (Dube and Nhamo, 2018). Magadza (2011) reported a declining trend in rainfall beginning in the early 1980's though other researchers did not find significant changes in Zambia's rainfall (Kampata et al., 2008; Mulenga et al., 2017; Stern and Cooper, 2011). Drought and seasonal floods have increased in Zambia and the worst drought was experienced in 1991/1992 (The Government of the Republic of Zambia et al., 2007). The latest drought was recorded in 2007/2008 rainy season (Bwalya, 2010). During the 1978/1979 season, Zambia experienced the wettest conditions ever (Bwalya, 2010). Projections show that by the year 2070, Zambia's rainfall will increase with reference to 2010 (The Government of the Republic of Zambia et al., 2007).

In Zambia, the potential effects of climate change on the forests remain uncertain and the response of net primary productivity (NPP) to climate change could be diverse due to strong heterogeneity and variability in regional contemporary climatic conditions and the differences in projected future climatic conditions. Thus, understanding how terrestrial NPP responds to climate change is important as it subsequently affects various ecosystem services (Piearce, 1986a; Piearce, 1986c; Sarmiento and Gruber, 2002). In this study, we applied the LPJ-GUESS model (Ahlström et al., 2012; Smith et al., 2001) to quantify the projected future effects of changes in temperature, rainfall, $CO_2$ concentration, solar radiation, and number of wet days on NPP

under RCP4.5 and RCP8.5. We projected changes in NPP for the end of the 21$^{st}$ century (2070-2099) with reference to 1960-1989 period as baseline. Our overall objective was to assess the future response of the NPP to climate change in the Zambezi teak forests along a rainfall gradient in Zambia.

## 2        Materials and methods

### 2.1 Study sites

We carried out the study for the Zambian Zambezi teak forests at the Kabompo (14° 00.551S, 023° 35.106E), Namwala (15° 50.732S, 026° 28.927E), and Sesheke (17° 21.278S, 24° 22.560E) sites. At the Sesheke site, the Masese forest reserve was assessed while at the Namwala site, we assessed the Ila forest reserve. At the Kabompo site, we studied the Kabompo and Zambezi forest reserves. While the Masese forest reserve is found in the drier agro-ecological zone I, the Kabompo and
Zambezi forest reserves are located in the wetter ecological zone II. The Ila forest reserve at the Namwala site stretches along ecological zones I and II (Fig. 1 and Table 1).

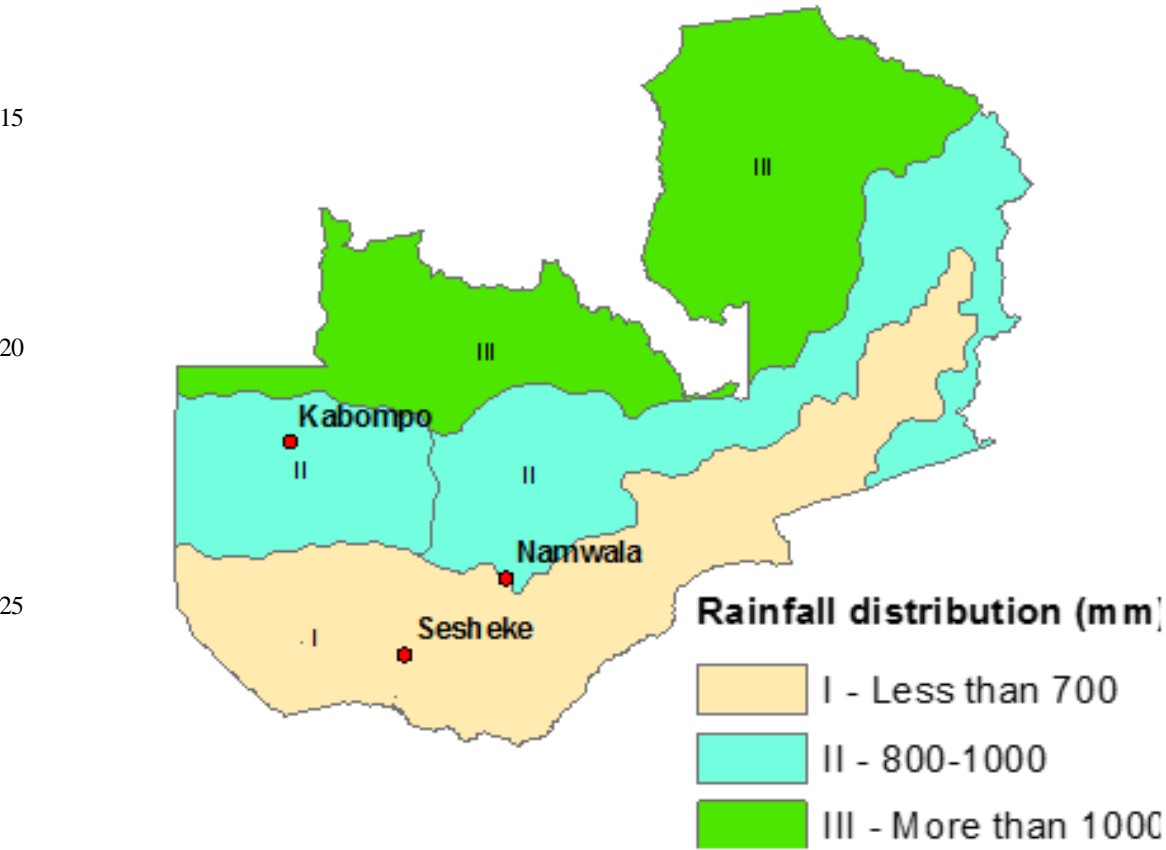


**Figure 1.** Distribution of rainfall and study sites following the ecological zones I, II, and III (Wamunyima, 2014)

**Table 1.** Climate and soil characteristics at Kabompo, Namwala, and Sesheke. For rainfall and temperature, the period covered for average values presented are given in brackets.

| Parameter | Kabompo | Namwala | Sesheke |
|---|---|---|---|
| Coordinates | 14°00.551S, 023°35.106E | 15°50.732S, 026°28.927E | 17°21.278S, 24°22.560E |
| Ecological zone | II | I and II | I |
| Total annual rainfall (mm) | 983 (1944-2011) | 905 (1944-2011) | 643 (1947-2011) |
| Mean annual temperature (°C) | 21.4 (1959-2003) | 21.6 (1959-2011) | 21.5 (1950-2011) |
| Nitrogen (%) | 0.04 | 0.03 | 0.03 |
| Clay (%) | 0.53 | 0.56 | 0.31 |
| Silt (%) | 0.54 | 0.55 | 0.43 |
| Fine sand (%) | 35.51 | 63.22 | 24.89 |
| Course sand (%) | 63.42 | 35.70 | 74.31 |
| pH-H2O | 5.55 | 5.74 | 5.86 |
| Organic carbon (%) | 0.77 | 0.73 | 0.90 |
| Soil bulky density (g/m³) | 1.54 | 1.53 | 1.87 |

## 2.2 Description of the Zambezi teak forests.

The Zambezi teak forests cover 9 % of Zambia's total forests' area (Matakala et al., 2015) and store between 15t C ha$^{-1}$ to 36t C ha$^{-1}$ (Ngoma et al., 2018a) across a south-north climatic gradient with annual rainfall ranging from 700 mm to 1100 mm. They are found on the flat areas covered with a thick layer of Kalahari sands (The Government of the Republic of Zambia, 1996). The forests are composed of 80 tree species (Ngoma et al., 2018a, b) but *Baikiaea plurijuga* Harms is most common (i.e. 50 % of the total surveyed stems) (Ngoma et al., 2018a, b; Ngoma et al., 2017). These forests are two storeyed with either

a closed or open canopy (Mulolwa, 1986). Trees of the Zambezi teak forests grow up to 20 m high and 120 cm in diameter (Piearce, 1986b) and they tolerate shade. For example, seedlings of *Baikiaea plurijuga* need some shade to survive (PROTA4U, 2017). Shade tolerant species are able to dominate a closed-forest and seeds are able to germinate in a closed forest. For *Baikiaea plurijuga*, regeneration is mainly from seeds, though seedlings are usually destroyed by wild animals within the forests (Piearce, 1986a). The forests have a deciduous shrub layer which is locally known as mutemwa and grows up to 3 m

to 6 m high. During the rainy season the forests have a ground layer of herbs and grasses (Mulolwa, 1986). These herbs and grasses have shallow root systems that develop during the rainy season and die or become dormant during the dry season. The Zambezi teak forests are threatened by deforestation, and between 1975 to 2005 the forests halved in area (Musgrave, 2016) due to logging and agricultural activities, driven by economic and population growth (Matakala et al., 2015; Theilade et al., 2001). Climate change is another threat to the Zambezi teak forests. Following the characteristics of the Zambezi teak forests

and the defined PFTs (Ahlström et al., 2012; Sitch et al., 2003), we used the "deciduous tropical broadleaved rain green" PFT in our study. Deciduous tropical trees shed their leaves during the dry season (See Appendix A in Ngoma et al. (2017) for the Zambezi teak forests in different seasons of the year).

## 2.3 Soil and tree parameter data sources

We collected data on soil and vegetation parameters from the field survey (Ngoma et al., 2018a, b). We analysed soil parameters down to 1.5 m depth from the plots where we conducted vegetation survey (Ngoma et al., 2018a). We determined soil texture and bulk density following the method by Sarkar and Haldar (2005) and organic carbon by Walkley and Black (1934) (See Supplementary Information Table S1 for details). Data on crown area, tree diameter, and total tree height were collected from the field survey in our previous studies (Ngoma et al., 2018a, b), while data on leaf longevity were determined from Specific Leaf Area (SLA) (Reich et al., 1997) to parameterize the LPJ-GUESS model. We determined SLA from the tree leaves we collected from the trees that we felled to develop Allometric equations (Ngoma et al., 2018a, b). Data on vegetation carbon and tree ring indices for the LPJ-GUESS model validation were taken from the biomass (Ngoma et al., 2018a, b) and dendrochronological (Ngoma et al., 2017) studies respectively.

## 2.4 Climate data sources

We used RCP4.5 and RCP8.5 with an ensemble of five Global Circulation Models (GCMs): CNRM-CM5, EC-EARTH, HADGEM2-ES, IPSL-CM5A-LR, and MPI-ESM-LR (See Supplementary Information Table S2 for full names). The climate data were re-gridded from the original spatial resolution of the climate model to a resolution of 0.5° x 0.5°. We applied the method by Piani et al. (2010) to bias-correct daily rainfall and temperature (minimum and maximum) values from the five GCMs against the WATCH Forcing Data (Weedon et al., 2011). The solar radiation data were bias-corrected following the method by Haddeland et al. (2012) using WATCH forcing data series (1971–2000) as a reference.

Both contemporaneously and projected data on temperature, rainfall, solar radiation and number of wet days were taken from CMIP5: CNRM-CM5.1 (Voldoire et al., 2013), EC-Earth (Hazeleger et al., 2011), HADGEM2-ES (Collins et al., 2011), IPSL-CM5A-LR (Dufresne et al., 2013), and MPI-ESM-LR (Giorgetta et al., 2016; Jungclaus et al., 2013). Data on $CO_2$ concentration were taken from Representative Concentration Pathway (RCP) database: RCP4.5 (Clarke et al., 2007; Smith and Wigley, 2006; Wise et al., 2009) and RCP8.5 (Riahi et al., 2007).

We collected local climate data from local weather stations. Forcing data on observed temperature, rainfall, and cloud cover were collected from local weather stations within the respective ecological zones. We collected local climate data from 15, 13, and 28 weather stations for Sesheke, Kabompo and Namwala sites respectively (See Supplementary Information Fig. S7). The surveyed Ila forest reserve at the Namwala site stretches in zones I and II, thus climate data were averaged from all local weather stations in both zones. Contemporaneously number of wet days were downloaded from Climatic Research Unit (CRU) website (University of East Anglia Climatic Research Unit et al., 2015)

## 2.5 Projected climate conditions: RCP4.5 and RCP8.5

In this study, we defined climate as the average weather pattern over a period of 30 years. Climate change was thus, defined as the difference between the climates of two periods. We used 1960-1989 as the baseline to determine the relative climate change for the end of the 21st century (2070-2099).

Data from CMIP5 shows that temperature (Fig. 2b) and incoming solar radiation (Fig. 2c) are projected to increase by the end of the 21st century (2070-2099) at all sites under both RCPs relative to 1960-1989. Temperature increases by 3 °C at all sites by the end of the 21st century under RCP4.5 while, under RCP8.5, temperature is projected to increase by 5 °C at the Kabompo and Namwala sites, and by 6 °C at the Sesheke site. Rainfall is projected to decrease by 33 mm and 23 mm at Kabompo and Sesheke respectively, and to increase by 28 mm at Namwala under RCP8.5 by 2099. Under RCP4.5, rainfall will increase by

32 mm and 3 mm at Namwala and Sesheke respectively while at Kabompo, rainfall will decrease by 10 mm by the end of the 21st century (Fig. 2a). The number of wet days will decrease at all sites under both RCPs by the end of the 21st century (Fig. 2d). Carbon dioxide concentration is projected to almost double under RCP8.5 by 2099 (Fig. 2e).

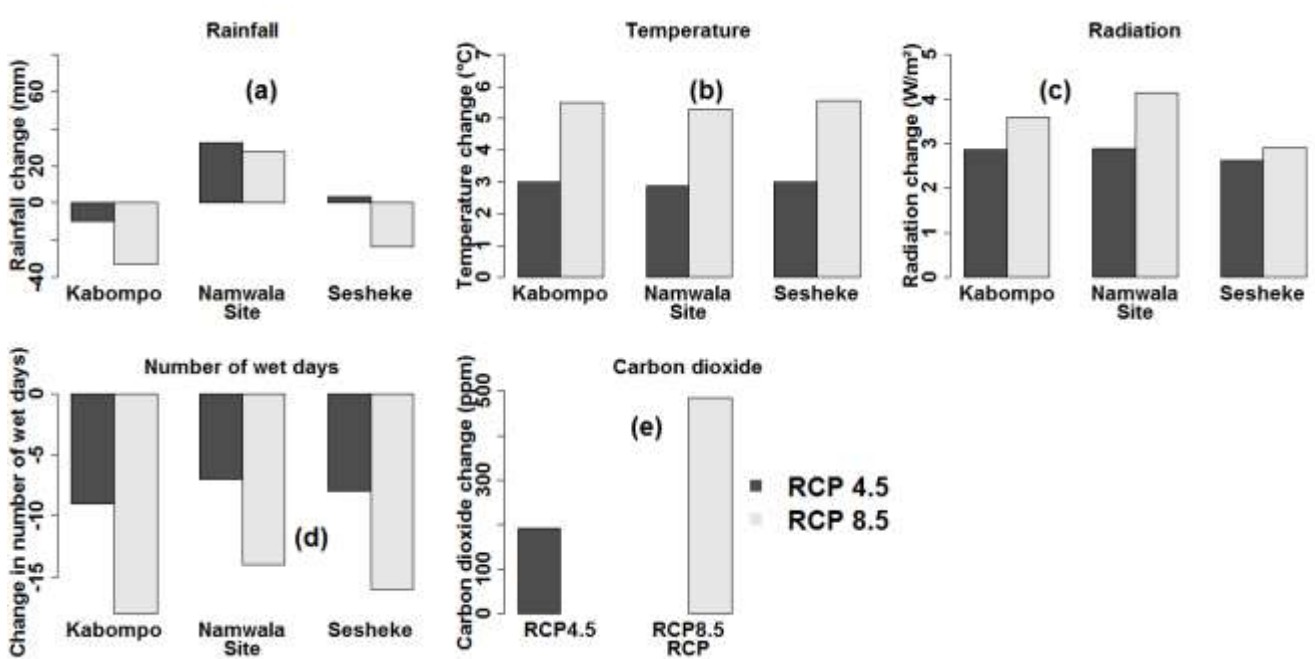

**Figure 2.** Projected changes in rainfall (a), mean temperature (b), incoming solar radiation (c), number of wet days (d), and $CO_2$ concentration (e) under RCP4.5 and RCP8.5 by the end of the 21st century. End of the 21st century is the period 2070-2099. Values were determined as
means of the five GCMs and changes were determined with reference to 1960-1989 period as baseline. For sources of data, refer to Sec 2.4.

## 2.6 The LPJ-GUESS model description

LPJ-GUESS (Ahlström et al., 2012; Smith et al., 2001) is a dynamic vegetation model (DVM) optimised for local, regional, and global applications. However, we applied the model at the local scale in our study. The model uses temperature,

precipitation, solar radiation, number of wet days, $CO_2$ concentrations, and soil texture as input variables to simulate the exchange of water and carbon between soils, plants, and the atmosphere. The ecosystem composition and structure is then determined for each simulated scale of which in our study, it was for local scale. One grid cell has a number of patches of approximately 0.1 ha in size (Smith et al., 2001). Each patch has a mixture of PFTs (Ahlström et al., 2012; Sitch et al., 2003),

distinguished by their bioclimatic niche (distribution in climate space), growth form (tree or herb), leaf phenology (evergreen, summer green, or rain green), photosynthetic pathway (C3 or C4), and life history type (shade-tolerant or shade-intolerant). In a patch, each woody plant belongs to one PFT and has a unique set of parameters that control establishment, phenology, carbon allocation, allometry, survival response to low light conditions, scaling of photosynthesis and respiration rates, and the limits in climate space the PFT can occupy. These parameters are represented in the model through different equations. The equations

given below show how some of the parameters that we modified from the default to local values (See Table 2) are represented in the model.

In LPJ-GUESS model, leaf longevity has a direct relationship with carbon storage. This relationship is implemented by relating the specific leaf area (SLA; $m^2$ $kg$ $C^{-1}$) to leaf longevity (See Eq. (1)) according to the 'leaf economics spectrum' (Reich et al., 1997).

$$SLA = 0.2 \times e^{(6.15 - 0.46 \times \ln(12\alpha))} \tag{1}$$

where α is leaf longevity (in years).

Photosynthesis, stomatal conductance, plant water uptake and evapotranspiration are modelled concurrently on a daily time step by a coupled photosynthesis and water module, which was adapted from the BIOME3 model (Haxeltine and Prentice, 1996). Soils have an upper (0.0 m to 0.5 m) and a lower (0.5 m to 1.5 m) layer, identical in texture. Water enters the upper soil

layer through precipitation. Transpiration and evapotranspiration deplete the water content of the soil. Additional depletion of soil water may occur through percolation beyond the lower soil layer and out of reach by plant roots. Uptake by plants is partitioned according to the PFT specific fraction of roots situated in each layer (Smith et al., 2001).

Net Primary Productivity (NPP) is determined from Gross Primary Productivity (GPP) after accounting for maintenance and growth respiration. The accrued NPP is allocated on an annual basis to leaves, sapwood and fine roots, enabling tree growth

(Sitch et al., 2003). This allocation is adjusted such that the following four allometric equations, or "constraints", controlling the structural development of the average individual, remain satisfied: Leaf area to sapwood cross-sectional area relationship (McDowell et al., 2002) (See Eq. (2)), the functional balance constraint (See Eq. (3), the stem mechanics equation (Huang et al., 1992) (See Eq. (4)), and the crowding constraint (See Eq. (5)) (Reineke, 1933). In LPJ-GUESS, crown area ($m^2$ per individual) is determined from stem diameter (See Eq. (6)) and tree diameter is derived from the sapwood, heartwood, and

wood density (See Eq. (7)). The reader is referred to Smith et al. (2001) for details

We used LPJ-GUESS version 3.0 and implemented a 'cohort mode' for our study (Braakhekke et al., 2017; Smith et al., 2001). Though this model version accounts for nitrogen dynamics in soil and vegetation, we did not switch nitrogen on during our simulations.

$$LAI = K_{lasa} \times SA \tag{2}$$

$$C_{leaf} = K_{lr} x \, \omega \times C_{root} \tag{3}$$

$$H = K_{allom2} \times D^{K_{allom3}} \tag{4}$$

$$N \approx D^{-k_{rp}} \tag{5}$$

$$CA = K_{allom1} \times D^{K_{rp}} \tag{6}$$

$$D = [\frac{4 \, x \, (C_{sapwood} + C_{heartwood})}{WD \times \pi \times K_{allom2}}]^{1/(2+K_{allom3})} \tag{7}$$

Where $K_{lasa}$, $K_{lr}$, $K_{rp}$, $K_{allom1}$, $K_{allom2}$, and $K_{allom3}$ are all constants, LAI is the leaf area index, SA is the sapwood cross section area (m²), $C_{leaf}$ is leaf carbon (kg C m²), $C_{root}$ is root carbon (kg C m²), $\omega$ is the mean annual value of a drought-stress factor which varies between 0 and 1 and higher values represent greater water availability. In our study we used a value of 0.35, which is the water stress threshold for leaf abscission (i.e. the point at which the leaves start shading). H stands for total tree height (m), D is tree diameter (m), N stands for population density (individuals per m²), CA is crown area (m²), WD stands for wood density (kg C m⁻³), $C_{sapwood}$ is sapwood carbon (kg C m²), and $C_{heartwood}$ is heartwood carbon (kg C m²).

## 2.7 Model set-up

We initiated the model with a 1000 year spin-up at each site to allow the model time to reach equilibrium in all carbon pools. We spun-up the model with observed climate data from local weather stations and contemporaneously modelled climate data during the respective model runs. Observed climate data are temperature, rainfall, and cloud cover data observed from local weather stations in the respective study sites, while contemporaneous data on $CO_2$ concentration were downloaded from the RCP database (RCP Database, 2018). Data on the number of wet days per month were downloaded from Centre for Environmental Data Analysis (University of East Anglia Climatic Research Unit et al., 2015). Contemporaneously modelled climate data are temperature, rainfall, number of wet days per month, and solar radiation averaged from the five GCMs described under Section 2.4, and $CO_2$ concentration data downloaded from RCP data base (RCP Database, 2018).

Using observed local climate data, we forced LPJ-GUESS during the spin-up with repeated cycle of 30-year climate data for 1959-1988 and a constant $CO_2$ concentration of 316 ppm, corresponding to the observed value for 1959. After the 1000-year spin-up period, the model was forced with a 53-year observed climate and $CO_2$ values, corresponding to the 1959-2011 period at Namwala and Sesheke sites. We forced the model with a 45-year observed climate and $CO_2$, corresponding to the 1959-2003 period at Kabompo site. $CO_2$ had reached 375 ppm and 390 ppm by 2003 and 2011 respectively.

Before forcing the model with projected climate data, we first spun-up the model with 30 years modelled climate data from 1960-1989 and a constant $CO_2$ of 317 ppm, corresponding to 1960. We then forced the model with 46-year contemporaneously modelled climate data for the period 1960-2005. We used $CO_2$ data for the same period of 1960-2005 and by 2005, $CO_2$ had reached 379 ppm.

After the spin-up period, and using observed local climate data at the respective sites as forcing, we performed a factorial experiment to determine the effects of various tree parameters (Table 2) and soil textures (Table 1 and Supplementary

Information Table S1) on different model output. We first ran the model with default tree parameters that were provided together with the model code (These are tree parameters from literature, but provided together with the model code. See Table 2). After identifying some limitations (Section 3.2), we tested the effects of local tree parameter values listed in Table 2 that coincided with the locations of our measurement plots (Ngoma et al., 2018a). We assessed effects of changing each parameter separately and of changing all parameters combined at each site (Table 2). We further assessed the effects of soil by running the model with default soil parameters (provided with the model code on a 0.5 x 0.5 global grid) and with local soil parameters derived from samples at the respective sites (Supplementary Information Table S1). Results at each site were averaged for 45 years (1959-2003) at Kabompo and for 53 years (1959-2011) at the Namwala and Sesheke sites. Forcing the model with observed climate data and using local tree and soil parameters, we compared the LPJ-GUESS simulated carbon stocks and NPP with measured carbon stock (Ngoma et al., 2018a, b) and tree-ring indices (Ngoma et al., 2017) respectively.

**Table 2.** Local and default tree parameter values used in LPJ-GUESS. $K_{rp}$, $K_{allom1}$, $K_{allom2}$, and $K_{allom3}$ are constants in allometric equations (See Sec 2. and Smith et al. (2001). Default parameters were provided together with the model code (Smith et al. (2001)).

| Site | $K_{allom1}$ | $K_{allom2}$ | $K_{allom3}$ | $K_{rp}$ | Maximum crown area ( m²) | Wood density (kg m⁻³) | Leaf longevity (Years) |
|------|--------|--------|--------|--------|--------|--------|--------|
| **Default** | 250 | 60 | 0.67 | 1.60 | 50 | 200 | 0.50 |
| **Kabompo** | 279 | 21 | 0.48 | 1.11 | 336 | 790 | 0.95 |
| **Namwala** | 424 | 20 | 0.56 | 1.39 | 269 | 790 | 0.94 |
| **Sesheke** | 480 | 31 | 0.58 | 1.19 | 452 | 790 | 0.94 |

We performed a factorial experiment for projected effects of temperature, rainfall, $CO_2$ concentration, incoming solar radiation, and number of wet days per month for the end of the 21st century (2070-2099) following RCP4.5 and RCP8.5. To isolate the contemporary effects of each of these climatic variables, the model was forced with the 1960-2005 values of the input climate variable of interest while keeping the 1960 values constant for the other input climatic variables. When assessing the projected effects, we forced the model with projected climate values for the period 2006-2099 of the input climate variable of interest, while keeping the 2006 value constant for the other input climatic variables.

## 3      Results

### 3.1     The LPJ-GUESS model validation

We forced the LPJ-GUESS model with observed local climate data and used local tree (Table 2) and soil parameter values (Supplementary Information Table S1) to validate the model. We validated the model by comparing standardised tree-ring indices to LPJ-GUESS simulated annual NPP, i.e. for the period 1970-2003 at the Kabompo site and 1959-2011 at the Namwala and Sesheke sites. The Nash–Sutcliffe model efficiency (NSE) coefficient indicated that the tree ring indices and LPJ-GUESS simulated NPP compared poorly at all the three sites (Kabompo: NSE = -2.2334, Namwala: NSE = -1.4555, and Sesheke: NSE = -2.0253).

We also validated the model by comparing measured vegetation carbon with simulated vegetation carbon at the respective study sites. We forced the model with local climate data and ran it with default soil and tree parameters to assess its performance and the model over-estimated vegetation carbon stock at all sites by between 44 % and 145 %. However, replacing default with local soil parameters (Supplementary Information Table S1), maximum crown area, wood density, leaf longevity, and

allometry (Table 2), the error reduced to 5 %, 47 %, and 17 % at the Kabompo, Namwala, and Sesheke sites respectively compared to measured vegetation carbon (Fig. 3).

We further assessed the LPJ-GUESS model performance by comparing measured and simulated tree heights and crown area. Using Eq. (4), tree heights estimated using default tree parameter values (Table 2) of $K_{allom2}$ and $K_{allom3}$ were taller than those estimated using local tree parameters of these same constants for the measured tree diameter at breast height (DBH) at

all sites (Fig. 4). Applying the Mean Absolute Percentage Error (Sileshi, 2014) to indicate allometric model performance, tree heights were over-estimated by 111 % at Kabompo, 156 % at Namwala, and 56 % at Sesheke sites when we used default tree parameter values of $K_{allom2}$ and $K_{allom3}$ in the allometric equation compared to measured tree heights. Using local tree parameter values (Table 2), tree heights were over-estimated by 2 % and 1 % at Kabompo and Namwala and under-estimated by 8 % at Sesheke respectively. Thus, both default and local tree parameters over-estimated tree heights at Kabompo and

Namwala compared to measured heights, though the over-estimation was largest with default parameters (Fig. 4).

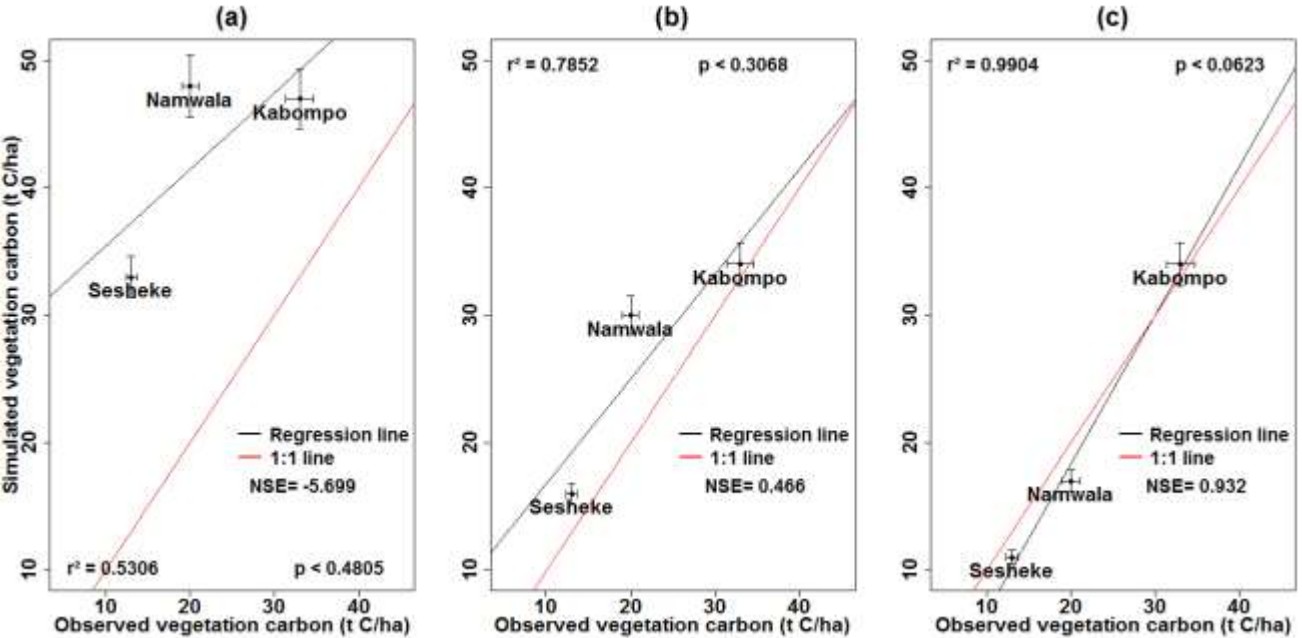

**Figure 3**. Measured versus LPJ-GUESS simulated vegetation carbon stock simulated with default soil parameters, default tree parameters, and observed local climate (a); local soil, local tree parameters, and observed local climate (b); and with local soil, local tree parameters, and modelled contemporaneously climate (c). NSE stands for Nash-Sutcliffe Efficiency (Nash and Sutcliffe, 1970)

The crown area, estimated with Eq. (6), was under-estimated by 61 % at Kabompo and Namwala and by 76 % at Sesheke when we used default tree parameters. However, with local tree parameters, the model under-estimated crown area by 15 %, 11 %, and 23 % at Kabompo, Namwala, and Sesheke, respectively compared to measured crown area (Fig. 5 and Table 2).

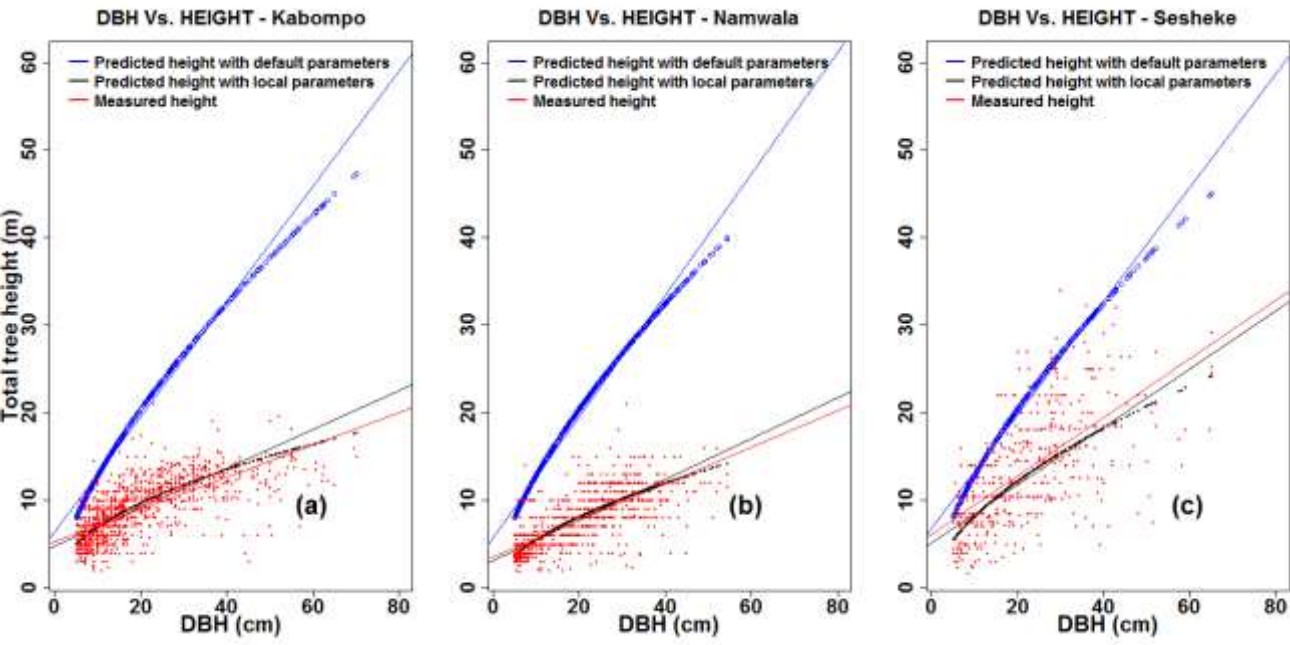

**Figure 4**. Measured and predicted total tree height, plotted against DBH at Kabompo (a), Namwala (b), and Sesheke (c).

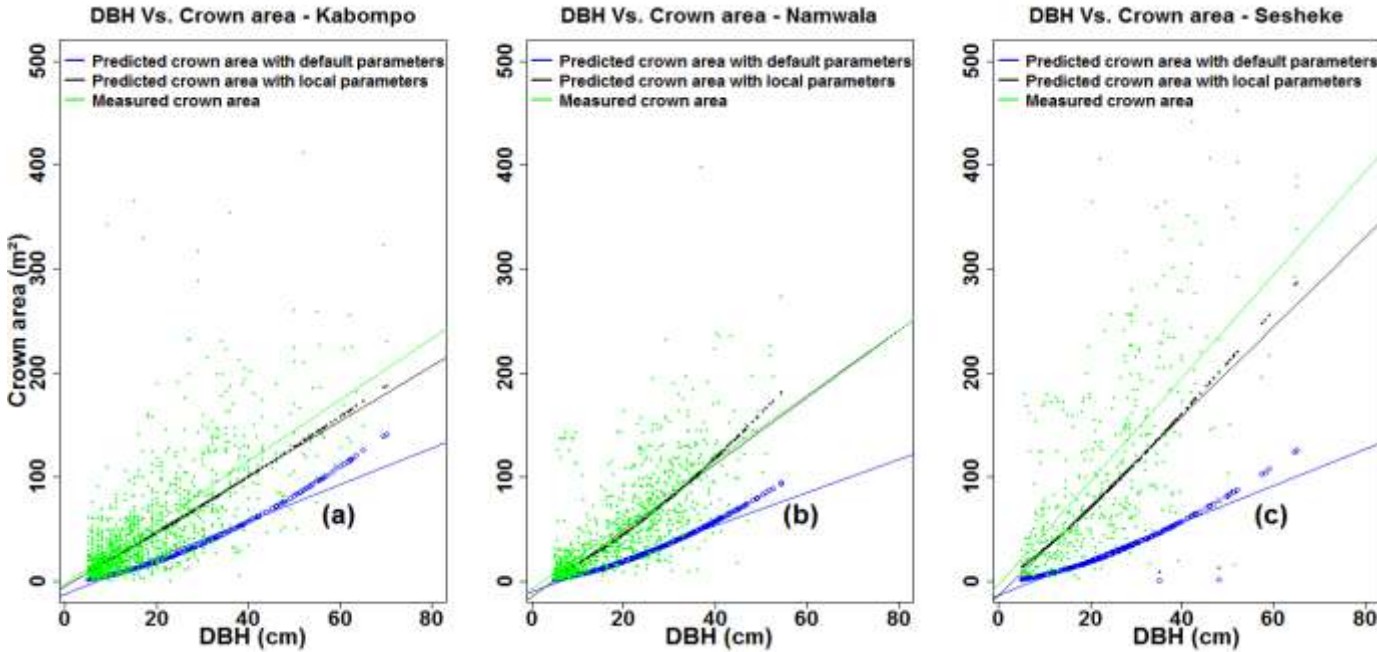

**Figure 5.** Measured and predicted crown area plotted against DBH at Kabompo (a), Namwala (b), and Seseheke (c).

### 3.2 Carbon stocks, LAI and NPP

Running the LPJ-GUESS model with local soil and tree parameters, and forcing it with local observed climate data for the
period 1960-2003, vegetation carbon stocks, and Leaf Area Index (LAI) were highest at Kabompo, and Seseheke had the lowest
values. The aggregated three carbon pools (vegetation, litter, and soil carbon) were highest at Kabompo and lowest at
Namwala. Vegetation carbon was lower when we forced the LPJ-GUESS model with contemporaneously modelled climate
data for the period 1960-2003 at all sites compared to the values simulated with observed local climate data (Fig. 6 and
Supplementary Information Fig. S6). Vegetation carbon stocks, LAI, and NPP simulated with both local soil and local tree
parameters, and forcing the model with local climate data were lower at all sites compared to values generated by default tree
and soil parameters (Fig. 6 and Supplementary Information Fig. S6).

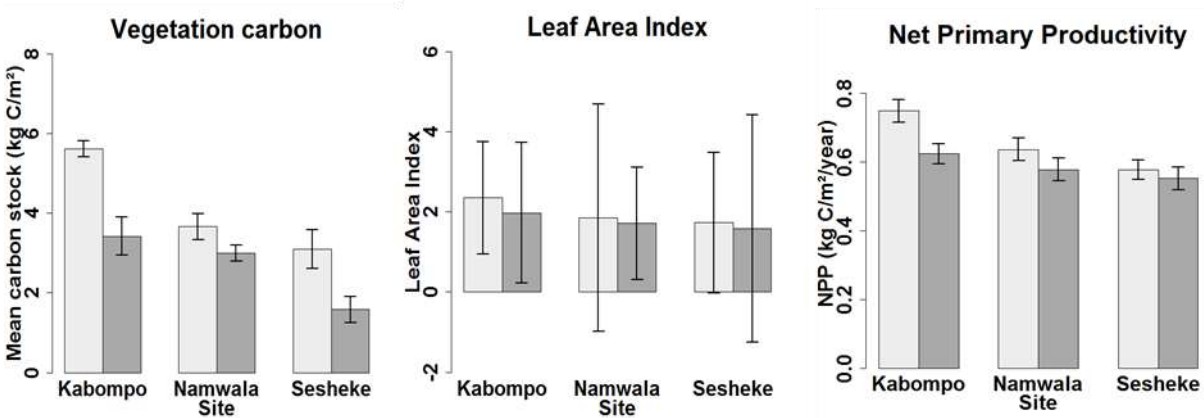

**Figure 6.** Mean annual vegetation carbon stocks, LAI and NPP simulated with local and default soil and tree parameter values, and forcing the model with local and modelled climate data. Simulations were done for the period 1959-2003. This figure only shows values simulated with a combination of default tree, default soil, and modelled climate data, and also a combination of local tree, local soil and local climate data. The reader is referred to supplementary information (Fig. S6) for the results of the effects of each of these default tree parameters, default soil parameters, local tree, local soil parameters, local climate, and modelled climate data.

### 3.3    Climate change effects on NPP

By the end of the 21st century, NPP is projected to increase at the Kabompo and Namwala sites but, reduces at the Sesheke site under both scenarios. NPP is projected to increase most at the Kabompo site under RCP8.5 (Fig. 7). Increased $CO_2$ concentration is projected to positively have most effects on NPP at Kabompo and Namwala under both RCPs, while under RCP8.5 decreased precipitation coupled with increasing temperature negatively affects NPP at Sesheke.

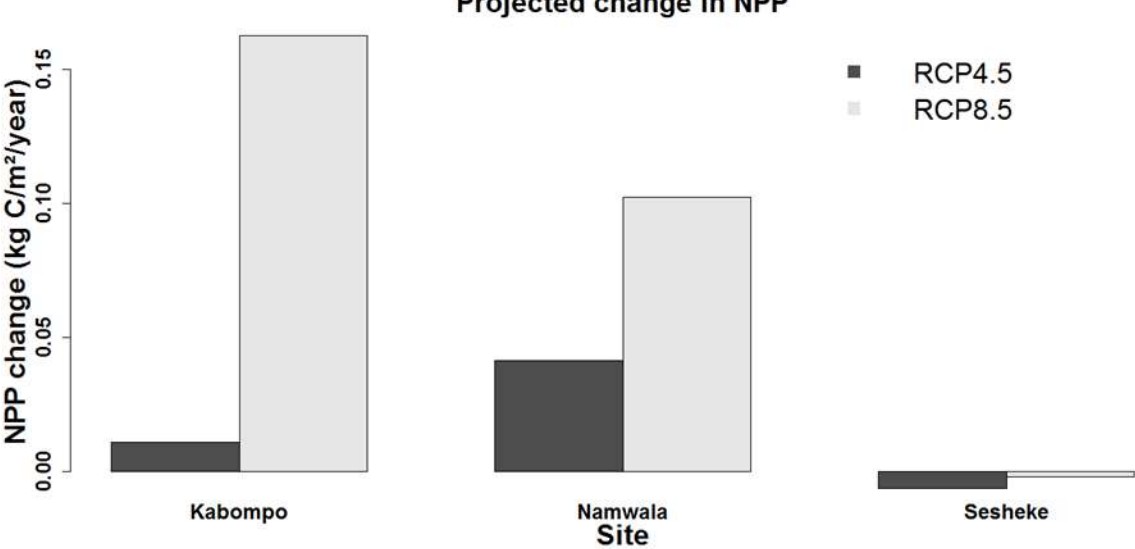

**Figure 7.** Projected changes in NPP at Kabompo, Namwala, and Sesheke resulting from combined changes in temperature, rainfall, $CO_2$ concentration, solar radiation and number of wet days by the end of the 21[st] century (2070-2099) with reference to 1960-1989 as the baseline.

## 4 Discussion

### 4.1 The LPJ-GUESS model performance

We generated new soil texture and tree parameter values for maximum crown area, wood density, leaf longevity, and allometry, and the results simulated with the LPJ-GUESS model improved when we used these local soil and tree parameter values compared to using the default parameters. The over-estimation of vegetation carbon that resulted from using default soil parameter values indicates the differences in clay, silt, and sand proportions between default and local soils of the Zambezi teak forests. Our field measurements (Ngoma et al., 2018a, b) showed that trees were between 2 m and 21 m tall. The high default tree heights of between 8 m and 47 m led to over-estimating vegetation carbon by between 33 % and 92 %.

LPJ-GUESS-simulated NPP and tree-ring indices compared poorly at all the three sites (Kabompo: NSE = -2.2334, Namwala: NSE = -1.4555, and Sesheke: NSE = -2.0253). This poor comparison is probably due to differences in the number of tree species incorporated in the two methods. We used one species only in the tree-ring analysis, while in modelling studies, which were conducted at ecosystem level, all available tree species in the forests were incorporated to determine the net NPP. The forests' survey that we conducted in 2014 (Ngoma et al., 2018a, b) showed that the Zambezi teak forests have eighty tree species. Thus, the net growth rate of these eighty species incorporated in the modelling studies is probably not the same as the growth rate of one dominant species used in tree-ring analysis. The total number of individual trees incorporated in tree ring

analysis and modelling studies also differed. While the model produced a mean NPP value from an ensemble of all trees in the studied forests, tree ring studies were conducted on a selected few trees. The trees from which NPP is generated represent a wide variability within the forests. For example, one tree may be restricted in its growth due to competitive pressure, while the overall NPP at the model's resolution includes the more successful trees within its estimates. However, the few trees

incorporated in the tree ring analysis represent few variability within the forests and results were generated from these few studied trees with either limited growth or successful growth compared to other trees in the forests.

Both, modelling and tree ring analysis showed significant positive spearman's correlations between productivity and rainfall of the previous two years at the Sesheke site (Supplementary Information Fig. S1 (i), and Fig. S2 (i)). These positive correlations between tree ring indices and rainfall (Supplementary Information Fig. S1 (i)), and also between LPJ-GUESS

simulated NPP and rainfall (Supplementary Information Fig. S2 (i)) of previous two years at Sesheke indicate a carry-over effect of rainfall on trees' productivity. Though rainfall of the previous years is probably captured by trees through soil moisture in the model, this aspect is not clearly addressed in LPJ-GUESS model. Babst et al. (2013) reported the lack of representation of carry-over effects of rainfall in Dynamic Global Vegetation Models (DGVM's). The clear representation of carry-over effects in LPJ-GUESS model would improve model results. Also, increasing the number of tree species in tree-ring analysis

would probably improve the results of the comparison between LPJ-GUESS simulated NPP and tree-ring indices. Thus, further tree-ring studies would need to be conducted with similar number of species as those included in modelling studies to validate the LPJ-GUESS model.

### 4.2     NPP's distribution

NPP was highest in the high rainfall receiving Kabompo site compared to the low rainfall receiving Sesheke site (Fig 6 and

Supplementary Information Fig. S6). The upward trend in NPP from the drier site to the wetter site was similar to the trend in LAI and vegetation carbon (Fig. 6 and Supplementary Information Fig. S6). The trend in NPP was also similar to the trend reported in literature where the forests growing in high rainfall receiving areas were more productive than the forests growing in arid regions (Cao et al., 2001; Delire et al., 2008; Ngoma et al., 2019; Williams et al., 2008).

### 4.3     NPP's climate response

We projected an NPP increase at Kabompo and Namwala caused by increasing $CO_2$ concentration and temperature. The positive temperature and $CO_2$ effects were clearly observed from the high positive spearman's correlations between NPP and temperature (See Supplementary Information Fig. S5) and NPP and $CO_2$ (See Supplementary Information Fig. S4). However, the positive temperature effects could just be up to an optimal temperature level. For tropical trees, carbon uptake reduces with leaf temperature of above 31 °C (Doughty and Goulden, 2008). Higher temperatures of above 31°C also reduce activities of

photosynthetic enzymes (Farquhar et al., 1980), resulting in reduced NPP.

The projected NPP increase at Kabompo and Namwala is in the same direction as the results reported by other researchers (Alo and Wang, 2008; Mohammed et al., 2018; Pan et al., 2015) for some parts of Africa (Table 3). Some modelling studies on tropical forests (Braakhekke et al., 2017; Ciais et al., 2009; Doherty et al., 2010; Melillo et al., 1993; Midgley et al., 2005; Pan et al., 2015; Thuiller et al., 2006) also reported large positive effects of increased $CO_2$ concentration on forests'

productivity. This positive effect could probably be due to increased Water-Use-Efficiency (WUE, which is a measure of a plant's water-use during photosynthesis in relation to the amount of water withdrawn (Grain Research and Development Cooperation, 2009)) by the plants. The stomata partially close to maintain a near constant concentration of $CO_2$ inside the leaf even under continually increasing atmospheric $CO_2$ levels. Such stomatal closure decreases evapotranspiration (Keenan et al., 2013) and thus increases WUE. The positive effects of increased $CO_2$ on NPP could also be due to increased Nitrogen-Use-

Efficiency (NUE, i.e., the amount of carbon converted into sugars during the photosynthetic process per unit of leaf nitrogen) (Davey et al., 1999). When $CO_2$ concentration increases, the amount of rubisco enzymes are reduced. As a consequence, foliar nitrogen is mobilized out of leaves and into other areas of the plant. This decreases the amount of nitrogen in the leaves. However, despite a reduction in leaf nitrogen, photosynthesis is still higher at elevated $CO_2$ concentrations. This result in increased carbon uptake at lower nutrient supplies. The higher photosynthesis activities and lower leaf nitrogen content

increase the photosynthetic NUE (Davey et al., 1999). However some other studies indicate that herbaceous plants and deciduous trees acclimate quickly to increased $CO_2$ concentrations by reducing photosynthetic capacity and stomatal conductance (Ellsworth, 1999; Mooney et al., 1999). As a result, the required water and nitrogen needed to fix a given amount of carbon is reduced (Chapin et al., 2007). However, such acclimation has sometimes no effect on the photosynthetic rate and stomatal conductance (Curtis and Wang, 1998). To what extent our modelling results are realistic is therefore not fully clear.

Currently, the responses of tropical trees and forests to increased $CO_2$ are still poorly understood (Thomas et al., 2008) since $CO_2$ enrichment experiments are lacking in the tropics. Such experiments should be done because they could explain whether the enhanced NPP that result from increased $CO_2$ is due to increased WUE, NUE or $CO_2$ fertilization. In our study, the spearman's correlations between tree ring indices and $CO_2$ concentration were not significant at all sites (Supplementary Information Fig. S3), contrary to modelling results, indicating the need for further research more especially the $CO_2$ enrichment

experiments to ascertain modelling results.

The projected decreased NPP under RCP8.5 at the Sesheke site results from high negative effects of the projected reduced rainfall coupled with increasing temperatures. NPP of the drier areas is mainly influenced by water by enhancing the WUE of vegetation (Yu and Chen, 2016). Reduced rainfall decreases soil water availability needed by the plants. High temperature enhances evapotranspiration resulting in reduced soil moisture (Miyashita et al., 2005). When soil water decreases, the stomata

close to restrict water loss. The closure of stomata prevents the movement of carbon into the plant, resulting in reduced NPP (McGuire and Joyce, 2005). Decreased soil water also limits nutrient absorption (e.g. Nitrogen) by the roots and transportation to the plants. Increased temperature enhances plant respiration, reducing photosynthetic activities (Burton et al., 2008; Wu et al., 2011). The projected reduced number of wet days likely have more effects on NPP at Sesheke under RCP4.5 by the year 2099. The projected NPP decrease at Sesheke is in the same direction as the findings of Delire et al. (2008) who reported an

NPP reduction of 12 % for the savanna forests by 2080. Similar results were also reported by Ngoma et al. (2019) who projected an NPP decrease of 8 % by the end of the 21st century for the whole of Africa. Furthermore, Alo and Wang (2008) projected NPP decrease in west and southern Africa.

The differences in NPP's response to climate change at each of the study sites is especially caused by variability in rainfall
and nutrient distribution (Fig. 1 and Table 1). Though the photosynthesis process is dependent on $CO_2$ concentration, plant's response to increasing $CO_2$ is limited by the availability of soil water and nutrients. Thus, plants growing in poor nutrient condition respond less to rising $CO_2$ concentration (Lloyd and Farquhar, 1996). This could be the case with the reduced NPP response at Sesheke where nitrogen content is lower than at Kabompo and Namwala (Table 1) despite the increasing projected $CO_2$ concentration. However, deciduous trees sometimes acclimate to increased $CO_2$ concentration by reducing photosynthetic
capacity and stomatal conductance (Ellsworth, 1999; Mooney et al., 1999). As a result, the required nitrogen and water needed to fix a given amount of carbon is reduced (Chapin et al., 2007), resulting in decreased NPP.

Generally, NPP would change at all the three studied sites (Kabompo, Namwala, and Sesheke) with the projected changes in climate and carbon dioxide concentration. However, the changes would fairly be small with the smallest changes recorded at the drier Sesheke site. This smallest change in NPP at the Sesheke site follows the smaller projected changes in rainfall (Fig.
2a).

**Table 3**. Projected changes in NPP: Current study compared to literature. A negative sign (-) under 'Change in NPP (%)', means a reduction in NPP.

| Change in NPP (%) | Forest biome | Study site | Period covered | Applied model | Reference | Comments |
|---|---|---|---|---|---|---|
| -16.98 | Tropical evergreen forest/woodland | Central and West Africa | 1950-2000 to 2070-2099 | IBIS | (Delire et al., 2008) | Used CRU data for control results |
| -24.18 | Tropical deciduous forest/woodland | | | | | |
| -6.06 | Savanna | | | | | |
| 10.00 | Grassland/steppe | | | | | |
| 0.00 | Open shrubland | | | | | |
| -50.00 | Desert | | | | | |
| -18.47 | Tropical evergreen forest/woodland | Central and West Africa | 1961-1990 to 2070-2099 | IBIS | (Delire et al., 2008) | Used climate data from Mark et al. (1999) for control results - Both rainfall and temperature changed |
| -26.03 | Tropical deciduous forest/woodland | | | | | |
| -15.12 | Savanna | | | | | |
| 12.99 | Grassland/steppe | | | | | |
| -6.78 | Open shrubland | | | | | |
| -16.67 | Desert | | | | | |
| 28.11 | All biomes | East Africa | 1981–2000 and 2080–2099 | LPJ DGVM | (Doherty et al., 2010) | Difference sources of climate data (Refer to the article) |
| -8 | All biomes | Whole Africa | 1950-2099 | Various models | (Ngoma et al., 2019) | Difference sources of climate data |
| **0.44** | **Deciduous forests** | **Kabompo - Zambia - Southern Africa** | **1960-1989 to 2070 - 2099** | **LPJ GUESS** | **Current study** | **RCP4.5** |

| 1.10 | Deciduous forests | Namwala - Zambia - Southern Africa | 1960-1989 to 2070 - 2099 | LPJ GUESS | Current study | RCP4.5 |
|------|-------------------|------------------------------------|--------------------------|-----------|---------------|--------|
| -0.04 | Deciduous forests | Sesheke - Zambia - Southern Africa | 1960-1989 to 2070 - 2099 | LPJ GUESS | Current study | RCP4.5 |
| 1.77 | Deciduous forests | Kabompo - Zambia - Southern Africa | 1960-1989 to 2070 - 2099 | LPJ GUESS | Current study | RCP8.5 |
| 0.69 | Deciduous forests | Namwala - Zambia - Southern Africa | 1960-1989 to 2070 - 2099 | LPJ GUESS | Current study | RCP8.5 |
| -0.01 | Deciduous forests | Sesheke - Zambia - Southern Africa | 1960-1989 to 2070 - 2099 | LPJ GUESS | Current study | RCP8.5 |

| Symbol | Meaning of symbol |
|--------|-------------------|
| LPJ-DGVM | Lund-Potsdam-Jena Dynamic Global Vegetation Model |
| IBIS | Integrated Biosphere Simulator |
| LPJ-GUESS | Lund-Potsdam-Jena General Ecosystem Simulator |

The different NPP responses to climate change at the three sites could also be attributed to differences in species composition and the variable responses of these distinct tree species to the environment caused by variation in their physiological properties. While 9 % of the total tree species are common in all the three sites, 25 % of the total surveyed species are found at Kabompo,
38 % at Namwala and 16 % at Sesheke only (Ngoma et al., 2018b).

We projected different NPP patterns at the three study sites using climate data from five GCMs, downscaled to 0.5° x 0.5° resolution. However, NPP projections depend on the accuracy of the climate data. It is therefore, worth to note that models are a simplification of the reality and are therefore associated with different uncertainties and assumptions. Uncertainties from GCMs increases with the downscaling of the climate results. Our NPP results were thus, affected by the uncertainties and
assumptions associated with these GCMs.

We carried out our study in the three study sites of the Zambezi teak forests in Zambia applying the LPJ-GUESS model. These sites experience some disturbances resulting from illegal activities (e.g. charcoal burning).The artificial disturbances are not captured by the model since the model does not provide for such kind of disturbances in the forests. Thus, an incorporation of such forest disturbances in the model would improve model results. The fires, which are also other forms of disturbances, are
common in the Zambezi teak forests. These fires are usually caused by humans during the dry season, and the LPJ-GUESS model does not provide for these artificial fires. The incorporation of these artificial fires would improve the model results further though more studies would need to be conducted to determine the frequency and intensity of these fires in the forests before incorporating them in the model. This would reduce the uncertainties of the model results.

Generally, there are some similarities in the results we generated in our study with literature (Tables 3) for similar forest types.
The differences in actual values hint at the differences in models applied and the extent of area coverage. For example, while we conducted our study at local level, other researchers conducted similar studies at regional level (Doherty et al., 2010). Studies conducted at regional level constitute average results of different biomes while our study covered one biome only at all the three sites. Other factors such as species composition and soils also differ between our study sites and study sites of

other researchers. We compared our results to few studies due to limited literature on modelling studies reported for African biomes. Also, studies using the same model as our study (LPJ- GUESS) are limited in Africa. We could not find any studies applying LPJ- GUESS model at local level in Africa as most studies are conducted at global level (Cao and Woodward, 1998; Schaphoff et al., 2006). Availability of such studies would give much insight on our results. This therefore presents an opportunity to focus modelling research in Africa so as to determine the potential response of the different biomes to climate change. However, our study highlighted the need to use local or regional specific parameter values in models in order to obtain reliable estimates unlike using default parameter values.

## 4.4    Conclusions

We generated new soil texture and tree parameter values for maximum crown area, wood density, leaf longevity, and allometry. Using these newly generated local parameters, we adapted and evaluated the dynamic vegetation model LPJ-GUESS for the historical climate conditions. The results simulated with the LPJ-GUESS model improved when we used these newly generated local parameters. This indicates that using local parameter values is essential to obtaining reliable simulations at site-level. The adapted model setup provided a baseline for assessing the potential effects of climate change on NPP in the Zambezi teak forests in Zambia. NPP was thus projected to increase by 1.77% and 0.69% at the wetter Kabompo, and by 0.44% and 0.10% at the intermediate Namwala sites under RCP8.5 and RCP4.5 respectively especially caused by the increased $CO_2$ concentration by the end of the 21st century. However, at the drier Sesheke site, NPP would respectively decrease by 0.01% and 0.04% by the end of the 21st century under RCP8.5 and RCP4.5. The projected decreased NPP under RCP8.5 at the Sesheke site results from the reduced rainfall coupled with increasing temperature. We thus demonstrated that differences in the amount of rainfall received in a site per year influence the way in which climate change would affect forests resources. The projected increase in $CO_2$ concentration would thus, have more effects on NPP in high rainfall receiving areas, while in arid regions, NPP would be affected more by the changes in rainfall and temperature. $CO_2$ concentrations would therefore be more important in forests that are generally not temperature or precipitation limited, however precipitation will continue to be the limiting factor in the drier site.

*Data availability*: Refer to sections 2.3 and 2.4 of this paper for sources of various data used in this article

*Author contribution*: Justine Ngoma, Bart Kruijt, Eddy Moors, Royd Vinya, and Rik Leemans conceived the idea and designed the study; Justine Ngoma prepared the paper, Justine Ngoma and Maarten C. Braakhekke analysed NPP data, Justine Ngoma and Iwan Supit analysed climate data; James H. Speer together with all other authors provided editorial comments, interpreted and discussed the results.

*Competing interests*: All authors have approved the final article and declare that there is no conflict of interest.

*Acknowledgements:* We would like to thank the Copperbelt University, the HEART project of the NUFFIC-NICHE programme, the International Foundation for Science (IFS) and the Schlumberger Foundation Faculty for the Future for providing financial support to conduct this research. We sincerely thank the LPJ-GUESS model development team at Lund University in Sweden for providing us with the model code and allowing us to use their model in our research.

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
