# Peer review of "Modelling the response of Net Primary Productivity of Zambezi teak forests to climate change along a rainfall gradient in Zambia"

_Biogeosciences, 2018_

## Referee Comment (RC1) · Anonymous Referee #1 · 14 Dec 2018

Ngoma and others model net primary productivity in Zambian teak forests in response to projected climate change. The results are interesting but the discussion is rather terse and multiple English usage issues will make it necessary to make substantial improvements before I can recommend the manuscript be published. I find the modeling analysis as a whole to be described well but had a difficult time trying to realize what I had learned from the analysis beyond what was already known about applying models at local scales. Emphasizing the novelty of the study and improving the discussion are necessary steps.

Minor comments: The introduction takes a rather conversational tone with rather obvi-

ous statements that don't need mentioning like 'Without a doubt, patterns of terrestrial NPP may respond to changes in climatic variables'. Of course they will, people know that plants respond to climate. These things need not be noted. Reference for 'Future climate trends for most of the Zambezian phyto-region point in the direction of increased aridity' Also citation for 'Further, during the past half a century, available evidence show that the temperature increased by 0.5 °C in Africa and minimum temperatures rose more than maximum temperatures'. What evidence? Does 'prediction is for a temperature rise by more than 3.4 oC above the 1981-2000 baseline' refer to maximum temperature, minimum, and average? The preceding sentence discussed minimum temperature. Here: 'Without a doubt, reduced rainfall coupled with an increase in temperature is known to suppress NPP in most terrestrial ecosystems' write instead 'Reduced rainfall couped with an increase in temperature suppresses NPP in most terrestrial ecosystems' 'Baikiaea Plurijuga', small p. A few too many significant digits in Table 1. The average total rainfall for example doesn't have 5 significant digits and many reported values are reported at a far higher degree of certainty than environmental measurements warrant. Page 7: why were these GCMs chosen? Figure 3 is not convincing; this doesn't validate the model if that is the goal of this analysis. The critique of the default parameters is nice. Figure 7 is very hard to read. I recommend trying a different way to display the data. I'm confused as to why 1.5 m soil depth is chosen if 'In the sites, trees access soil water down to more than 5 m depth according to the trees' rooting depth in the Zambezi teak forests.' This really doesn't make sense to me, even if your measurements suggest that 1.5 is the maximum depth at the sites, it is easy to underestimate rooting depth. What does this mean 'This opens the novel concept to improve and validate LPJ-GUESS model.'. Is this what the subsequent analysis is doing? Please re-read the manuscript carefully for usage, for example 'As a results,' on page 17. And the next sentence, 'to what extent modelling results are realistically since'. 'Activity of photosynthetic enzymes also reduces (Farquhar et al., 1980)'. Lots of wording needs changes. The Discussion was rather short. What sorts of uncertainties need to be reduced, what directed studies would improve results, what

have we learned from this study?

---

## Referee Comment (RC2) · Anonymous Referee #2 · 3 Jan 2019

The manuscript is straightforward and well written, and the research objectives are clearly defined, specifically projecting climate change effects on the Zambezi teak forest productivity along a rainfall gradient in southern Africa. The results are interesting; the authors projected a decrease in NPP at a drier site as a consequence of reduced rainfall with high temperature while the converse was in other two wetter sites in which the NPP is projected to be controlled by temperature and CO2 concentration. It is noted that similar studies had been conducted at regional level, this study at the local scale is new. I found that some discussions are lacking to justify the significance of the study and I will point out some minor comments which I feel could be improved.
1. The title could be improved again in my opinion; I found it a bit misleading as it sounds more like observational study than modelling study.

2. Page 3: The introduction doesn't describe much about the area of need for this study which I found it difficult to convince the readers the importance of this study. The authors state that Zambezi forests play a substantial role in mitigating climate change on line 24-25, but didn't elaborate further on this. I feel it is better to describe in details about the Zambezi forest in relation to NPP particularly the forest extent and carbon storage and also deficiency in the existing literature.

3. What are the uncertainties of projected changes in climate and NPP? I recommend to add error bars to the Figures 2, 7 and 8.

4. Page 6, line 28: LAI is a unitless measure.

5. Page 6, line 32: CAmax is not found in the listed equations.

6. Page 10, line 8-9: The authors described how much the rainfall will increase or decrease under RCP 8.5 but not for RCP 4.5. Please also provide values or statistics for RCP 4.5.

7. Page 16, line 23 to Page 17, line 7: I found these arguments or discussions are ambiguous and obscure. I don't understand what the authors mean by 'limited amount of soil water availability in LPJ-GUESS model'. Since the authors also discussed that the carry-over effects of rainfall on trees' productivity has been reported by other researchers, how does this be novel though?

8. Page 17, line 19-20: Please fix the typing error for 'As a result. . .'.

9. Page 18-19: Some acronyms are not found in Table 3 – JULES, ORCHIDEE, CEVSA, DLEM.

10. Page 19, line 3: Please fix the typing error. . . .in there physiological properties.

---

## Author Comment (AC1) · 23 Jan 2019

Thank you for your valuable comments. We incorporated suggestions from the reviewer and we also spent some time and efforts to improving the language in the revised version. We have attached a table under 'Fig 1 to Fig 4' where we have outlined our responses to the reviewer and the changes we made to the manuscript.

**Page 1 - Table 1: Responses to referee #1**

| Section of the paper | Comment from the referee #1 | Author's response | Author's changes in manuscript |
|---|---|---|---|
| General comments covering all sections of the paper | The results are interesting but the discussion is rather terse and multiple English usage issues will make it necessary to make substantial improvements before I can recommend the manuscript be published. I find the modelling analysis as a whole to be described well but had a difficult time trying to realize what I had learned from the analysis beyond what was already known about applying models at local scales. Emphasizing the novelty of the study and improving the discussion are necessary steps. | We acknowledge the concerns raised by the reviewer | • We improve the language in the revised version
• We improved the discussion and emphasized the novelty of the study. |
| Introduction | The introduction takes a rather conversational tone with rather obvious statements that don't need mentioning. | We acknowledge the concerns raised by the reviewer and we revised the introduction | The introduction was replaced with the new one. In this new introduction, we described in details about the Zambezi forests and included information on the forest extent, carbon storage, and deficiency in the existing literature. |
| Introduction | Baikiaea Plurijuga', small p | We revised the word 'Plurijuga' | We replaced capital 'P' with small 'p' |
| Materials and methods | A few too many significant digits in Table 1 | We revised the values in table 1 to reduce the number of significant digits. | Rainfall values were rounded to whole numbers and temperature values were written to one decimal place. |
| Materials and methods | Page 7: why were these GCMs chosen | We acknowledge the concerns raised by the reviewer and we revised the paragraph | We provided the reasons in the manuscript for our choice of GCMs |

**Fig. 1.** Page 1- Table 1 - Responses to referee #1

**Page 2 - Table 1 continues**

| Section of the paper | Comment from the referee #1 | Author's response | Author's changes in manuscript |
|---|---|---|---|
| Results | Figure 3 is not convincing; this doesn't validate the model if that is the goal of this analysis | We acknowledge the concerns raised by the reviewer | Figure 3 was deleted from the revised manuscript |
| Results | Figure 7 is very hard to read. I recommend trying a different way to display the data. | We acknowledge the concerns raised by the reviewer and we displayed the data in a different way that is easier for the reader to understand. | • This figure showed mean annual vegetation carbon stocks, LAI and NPP simulated with local and default soil and tree parameter values, and forcing the model with local and modelled climate data. In the manuscript, we only showed values simulated with a combination of default tree, default soil, and modelled climate data, and also a combination of local tree, local soil and local climate data. The results of the effects of each of these default tree parameters, default soil parameters, local tree, local soil parameters, local climate, and modelled climate data were taken to supplementary information (Figure S6).
 • After the revisions, figure 7 in the old version became figure 6 in the revised manuscript. |

**Fig. 2.** Page 2- Table 1 - Responses to referee #1

**Page 3- table 1 continues**

| Section of the paper | Comment from the referee #1 | Author's response | Author's changes in manuscript |
|---|---|---|---|
| Discussion | I'm confused as to why 1.5 m soil depth is chosen if 'In the sites, trees access soil water down to more than 5 m depth according to the trees' rooting depth in the Zambezi teak forests.' This really doesn't make sense to me, even if your measurements suggest that 1.5 is the maximum depth at the sites, it is easy to underestimate rooting depth | We acknowledge the concerns raised by the reviewer and revised our argument after gaining more insight from literature on how rooting depth affect water uptake by plants. | We removed this argument from our discussion after gaining more insight from literature on how rooting depth affect water uptake by plants. Literature (For example Christoffersen et al. (2014)) indicates that water uptake by plants is dependent on different factors and rooting depth is just one them. However, there is no direct relationship between water uptake by plants and the rooting depth. So far, no study has been conducted in the Zambezi teak forests to determine the depth at which the trees take up water.
The effect of rooting depth on water uptake by plants differs with locality and species (Christoffersen et al., 2014). Our previous studies (Ngoma et al., 2018a, b) reported different species composition at each of the three studied sites (Kabompo, Namwala and Sesheke), though some of them are common. Roots were only uprooted at the drier Sesheke site, indicating that the rooting depths of trees at the Kabompo and Namwala sites are not yet known. Thus, simulating tree growth using the default 1.5 m rooting depth was logical as we did not have full information on the rooting depth of trees at the other two sites (Kabompo and Namwala). However, we studied soil characteristics down to 1.5 m depth at all the three sites, giving us the needed soil information at all the sites. Thus, using the default and uniform 1.5 m rooting depth enabled us to easily compare results at the three sites. |

**References**

Christoffersen, B. O., Restrepo-Coupe, N., Arain, M. A., Baker, I. T., Cestaro, B. P., Ciais, P., Fisher, J. B., Galbraith, D., Guan, X., Gulden, L., van den Hurk, B., Ichii, K., Imbuzeiro, H., Jain, A., Levine, N., Miguez-Macho, G., Poulter, B., Roberti, D. R., Sakaguchi, K., Sahoo, A., Schaefer, K., Shi, M., Verbeeck, H., Yang, Z.-L., Araújo, A. C., Kruijt, B., Manzi, A. O., da Rocha, H. R., von Randow, C., Muza, M. N., Borak, J., Costa, M. H., Gonçalves de Gonçalves, L. G., Zeng, X., and Saleska, S. R.: Mechanisms of water supply and vegetation demand govern the seasonality and magnitude of evapotranspiration in Amazonia and Cerrado, Agric. For. Meterol., 191, 33-50, 2014.
Ngoma, J., Moors, E., Kruijt, B., Speer, J. H., Vinya, R., Chidumayo, E. N., and Leemans, R.: Below and above-ground carbon distribution along a rainfall gradient. A case of the Zambezi teak forests, Zambia  Acta Oecologica 87, 45-57, 2018a.
Ngoma, J., Moors, E., Kruijt, B., Speer, J. H., Vinya, R., Chidumayo, E. N., and Leemans, R.: Data for developing allometric models and evaluating carbon stocks of the Zambezi Teak Forests in Zambia, Data in Brief 17, 1361-1373, 2018b.

**Fig. 3.** Page 3- Table 1 - Responses to referee #1

**Page 4 - Table 1 continues**

| Section of the paper | Comment from the referee #1 | Author's response | Author's changes in manuscript |
|---|---|---|---|
| Discussion | What does this mean 'This opens the novel concept to improve and validate LPJ-GUESS model'. | We adjusted this sentence to make it easier for the reader to understand | We clarified in the revised manuscript that the clear representation of carry-over effects in LPJ-GUESS model would improve model results. We therefore removed the sentence 'This opens the novel concept to improve and validate LPJ-GUESS model' from the revised manuscript. |
| Discussion | Please re-read the manuscript carefully for usage, for example 'As a results,' on page 17. And the next sentence, 'to what extent modelling results are realistically since' | The sentences were revised | • The letter 's' was removed from the word 'results'. The correct word is 'result'.
• The sentence, 'to what extent modelling results are realistically since', was re-written as 'The reality of modelling results are therefore not certain since $CO_2$ enrichment experiments are lacking in the tropics' |
| Discussion | 'Activity of photosynthetic enzymes also reduces (Farquhar et al., 1980)'. Lots of wording needs changes. | The sentence was revised | The sentence was re-written as 'Higher temperatures of above 31°C also reduce activities of photosynthetic enzymes' |
| Discussion | The Discussion was rather short. What sorts of uncertainties need to be reduced, what directed studies would improve results, what have we learned from this study? | We acknowledge the concerns raised by the reviewer and we revised the discussion | We expanded the discussion by including the uncertainties that need to be reduced and by recommending some studies that would improve results. We further highlighted the lessons learned from the study by emphasising the novelty of the study |

**Fig. 4.** Page 4- Table 1 - Responses to referee #1

---

## Author Comment (AC2) · 23 Jan 2019

Thank you for your valuable comments. We incorporated suggestions from the reviewer and we also spent some time and efforts to improving the language in the revised version. We attached under Fig 1 to Fig 3 our responses to the reviewer and the changes we made to the manuscript.

[Figure]

Page 1 - Table 1: Responses to referee #2

| Section of the paper | Comment from the referee #2 | Author's response | Author's changes in manuscript |
|---|---|---|---|
| Title | The title could be improved again in my opinion; I found it a bit misleading as it sounds more like observational study than modelling study. | We modified the title to reflect the modelling approach. | The new title is: "Modelling the response of Net Primary Productivity of Zambezi teak forests to climate change along a rainfall gradient in Zambia" |
| Introduction | Page 3: The introduction doesn't describe much about the area of need for this study which I found it difficult to convince the readers the importance of this study. The authors state that Zambezi forests play a substantial role in mitigating climate change on line 24-25, but didn't elaborate further on this. I feel it is better to describe in details about the Zambezi forest in relation to NPP particularly the forest extent and carbon storage and also deficiency in the existing literature. | We acknowledge the concerns raised by the reviewer and we revised the introduction | The introduction was replaced with the new introduction. In this new introduction, we described in details about the Zambezi forests and included information on the forest extent, carbon storage, and deficiency in the existing literature. |
| Results | What are the uncertainties of projected changes in climate and NPP? I recommend to add error bars to the figures 2, 7 and 8. | We acknowledge the concerns raised by the reviewer and added the error bars as suggested | We added error bars to figures 2, 7 and 8 as highlighted by the reviewer. However, these figures 2, 7 and 8 in the old manuscript will be figures 2, 6 and 7 respectively in the revised manuscript |
| Materials and methods | Page 6, line 28: LAI is a unit less measure | We acknowledge the concerns raised by the reviewer and revised line 28 | The units were removed from LAI |
| Materials and methods | Page 6, line 32: CAmax is not found in the listed equations | We acknowledge the concerns raised by the reviewer and we revised line 32. | $CA_{max}$ was removed since it is not in the listed equations |

**Fig. 1.** Page 1- Table 1 - Responses to referee #2

**Page 2 - Table 1 continues**

| Section of the paper | Comment from the referee #2 | Author's response | Author's changes in manuscript |
|---|---|---|---|
| Results | Page 10, line 8-9: The authors described how much the rainfall will increase or decrease under RCP 8.5 but not for RCP 4.5. Please also provide values or statistics for RCP 4.5. | We acknowledge the concerns raised by the reviewer and we revised line 8-9. | The values of projected rainfall changes under RCP 4.5 were provided in the revised manuscript. |
| Discussions | Page 16, line 23 to Page 17, line 7: I found these arguments or discussions are ambiguous and obscure. I don't understand what the authors mean by 'limited amount of soil water availability in LPJ-GUESS model'.

Since the authors also discussed that the carry-over effects of rainfall on trees' productivity has been reported by other researchers, how does this be novel though? | We acknowledge the concerns raised by the reviewer and we revised this section | • The argument presented on page 16, line 23 to Page 17, line 7 has been removed from the discussion after gaining more insight from literature on how rooting depth affect water uptake by plants. Literature (For example Christoffersen et al. (2014)) indicates that water uptake by plants is dependent on different factors and rooting depth is just one them. However, there is no direct relationship between water uptake by plants and the rooting depth. So far, no study has been conducted in the Zambezi teak forests to determine the depth at which the trees take up water. The effect of rooting depth on water uptake by plants differs with locality and species (Christoffersen et al., 2014). Our previous studies (Ngoma et al., 2018a, b) reported different species composition at each of the three studied sites (Kabompo, Namwala and Sesheke), though some of them are common. Roots were only uprooted at the drier Sesheke site, indicating that the rooting depths of trees at the Kabompo and Namwala sites are not yet known. Thus, simulating tree growth using the default 1.5 m rooting depth was logical as we did not have full information on the rooting depth of trees at the other two sites (Kabompo and Namwala). However, we studied soil characteristics down to 1.5 m depth at all the three sites, giving us the needed soil information at all the sites. Thus, using the default and uniform 1.5 m rooting depth enabled us to easily compare results at the three sites.
• We clarified in the revised manuscript that the clear representation of carry-over effects in LPJ-GUESS model would improve model results. We therefore removed the sentence 'This opens the novel concept to improve and validate LPJ-GUESS model' from the revised manuscript |

**References**

Christoffersen, B. O., Restrepo-Coupe, N., Arain, M. A., Baker, I. T., Cestaro, B. P., Ciais, P., Fisher, J. B., Galbraith, D., Guan, X., Gulden, L., van den Hurk, B., Ichii, K., Imbuzeiro, H., Jain, A., Levine, N., Miguez-Macho, G., Poulter, B., Roberti, D. R., Sakaguchi, K., Sahoo, A., Schaefer, K., Shi, M., Verbeeck, H., Yang, Z.-L., Araújo, A. C., Kruijt, B., Manzi, A. O., da Rocha, H. R., von Randow, C., Muza, M. N., Borak, J., Costa, M. H., Gonçalves de Gonçalves, L. G., Zeng, X., and Saleska, S. R.: Mechanisms of water supply and vegetation demand govern the seasonality and magnitude of evapotranspiration in Amazonia and Cerrado, Agric. For. Meteorol., 191, 33-50, 2014.
Ngoma, J., Moors, E., Kruijt, B., Speer, J. H., Vinya, R., Chidumayo, E. N., and Leemans, R.: Below and above-ground carbon distribution along a rainfall gradient. A case of the Zambezi teak forests, Zambia  Acta Oecologica 87, 45-57, 2018a.
Ngoma, J., Moors, E., Kruijt, B., Speer, J. H., Vinya, R., Chidumayo, E. N., and Leemans, R.: Data for developing allometric models and evaluating carbon stocks of the Zambezi Teak Forests in Zambia, Data in Brief 17, 1361-1373, 2018b.

**Fig. 2.** Page 2- Table 1 - Responses to referee #2

[Figure]

**Page 3 - Table 1 continues**

| Section of the paper | Comment from the referee #2 | Author's response | Author's changes in manuscript |
|---|---|---|---|
| Discussions | Page 17, line 19-20: Please fix the typing error for 'As a result: : :'. | We corrected the typing error | We removed the letter 's' from the word 'results'. The correct word was 'result' |
| Discussions | Page 18-19: Some acronyms are not found in Table 3 – JULES, ORCHIDEE, CEVSA, DLEM. | All acronyms that are not found in table 3 were removed | We removed JULES, ORCHIDEE, CEVSA and DLEM. from the list of acronyms |
| Discussions | Page 19, line 3: Please fix the typing error. : : :in there physiological properties. | The typing error was corrected | The word 'there' was replaced with the word 'their' |

**Fig. 3.** Page 3- Table 1 - Responses to referee #2

---

## Referee Report (RR1)

[referee-annotated manuscript omitted]

---

## Author Response (AR2)

**Responses to anonymous referee # 1**

Thank you for your valuable comments. We incorporated suggestions from the reviewer and we also spent some time and efforts to improve the language in the revised version. Below are our responses to the reviewer and the changes we made to the manuscript. Changes in the revised manuscript are highlighted in yellow.

5  We are requesting the reviewer to read the supplementary information as well where it has been referred to in the manuscript. Some of the concerns especially on statistical tests are already addressed in the supplementary material.

| Main-section of the paper | Sub-section of the paper | Line number (Page No.) | Comment from the referee #1 | Author's response | Author's changes in manuscript |
|---|---|---|---|---|---|
| Not indicated | Not indicated | Not indicated | The discussion of default vs. local parameters for the model is unnecessary. In my opinion, the model should not be run with default values, and the presentation of those results does not add value to this manuscript. | We disagree with the reviewer on the suggestion not to report results generated with default parameter values. Documentation on LPJ-GUESS model (Ahlström et al., 2012; Sitch et al., 2003; Smith et al., 2001) indicate that global vegetation is categorised into ten (10) plant functional types (PFT). This means that any vegetation that is studied using the LPJ GUESS model (following these PFT) is expected to fall under any of these 10 PFT. Each PFT has unique parameter values that are uniform globally. These parameter values were generated based on studies conducted at global level. These are the values that we termed as 'default' in our study. Following the characteristics of the trees of the Zambezi teak forests, these forests fall under the 'Tropical broadleaved rain green' plant functional type. Thus, before using the local parameter values, it was important for us to know the results generated by the default tree and soil values. The different values generated using default and local parameter values indicated the error that would have been attributed by the model had we used only the default parameter values to generate our results. | We did not make any changes to the manuscript. |

| Main-section of the paper | Sub-section of the paper | Line number (Page No.) | Comment from the referee #1 | Author's response | Author's changes in manuscript |
|---|---|---|---|---|---|
| | | | | Our varying results generated using default and local parameter values also indicated how the model can be improved by using local parameter values. We therefore found it necessary to report results generated using both default and local parameter values. | |
| Not indicated | Not indicated | Not indicated | A small but substantial concern is in the validity of the NPP change results. I am not an expert on this model, and I do not understand specifically what the margin of error is for NPP estimates. It seems to me that the model estimates fairly small changes in NPP, which may be within the margin of error for the model, and therefore statistically insignificant. I would like to see this rigorously addressed in the methods, results, and discussion. | We agree with the reviewer that changes in NPP are fairly small. However, we determined the relative changes in NPP after simulating NPP values using LPJ-GUESS model. NPP values in the model are determined by the values of input parameters (temperature, rainfall, incoming solar radiation, carbon dioxide concentration, number of wet days, and soil texture) and not by the model. Thus the small values in NPP change simply indicates smaller effects of changes in input parameters since the NPP values are sensitive to changes in these input parameters. Relating to margin of error, LPJ-GUESS model does not give the margin of error. Thus, we could not report the margin of error from the model as requested by the reviewer. The accuracy of the model was determined by validating it with measured values. In our study we compared simulated to measured values of various parameters which included vegetation carbon. The reviewer can refer to the section 'The LPJ-GUESS model validation' in the manuscript | We addressed the possible causes of small changes in NPP in the discussion |
| Not indicated | Not indicated | Not indicated | I am curious about the use of terms such as 'correlation' in the absence of statistical tests and test statistics (results). Perhaps this is nit-picking, but I suggest avoiding these terms as they can be misconstrued from a statistical perspective. | Where we used the term 'correlations', we reported statistical tests and test results and most of these results are reported in the supplementary information. We have referred the reader to this supplementary information in our manuscript. We are therefore requesting the reviewer to read the supplimentantay | We did not make any changes to the manuscript |

| Main-section of the paper | Sub-section of the paper | Line number (Page No.) | Comment from the referee #1 | Author's response | Author's changes in manuscript |
|---|---|---|---|---|---|
| | | | | information as well for him/her to have full understanding of our results. | |
| Not indicated | Not indicated | Not indicated | I have noted other small editorial concerns such as the appropriate use of particular terms, and made suggestions for wording that I would consider more appropriate. | We acknowledge the concerns raised by the reviewer. | We revised sentences and words where the reviewer made some suggestions. |
| Abstract | Abstract | Line 7 (page 2) | Poor English in this sentence | We acknowledge the concerns raised by the reviewer and revised the sentence | We removed the word 'thus' from the sentence |
| abstract | Abstract | Line 10-12 (page 2) | This may go without saying. | It is not always that results improve after applying local parameter values. It was therefore important to emphasise on the improvements in the model results after applying local parameters values | The sentence was not changed |
| Abstract | Abstract | Line 17-18 (Page 2) | English should be improved in this sentence for clarity. | We acknowledge the concerns raised by the reviewer and revised the sentence. | We rephrased the sentence for clarity |
| Introduction | Introduction | Line 2 (page 3) | Are these multiple forest types that you are discussing here? This might deserve a sentence to explain that you are investigating several forest complexes, not just a single forest type | We were investigating three forest types differentiated by the annual amounts of rainfall received per year. In the introduction, we just provided general information about the Zambezi teak forests in the region. However, more details on the specific forests where we focused our study were provided in the 'methods and materials section'. | We added a sentence in the revised manuscript to indicate the distribution of these forests in southern Africa. However, the reviewer can find more details on the specific forests where we focused our studies on in the 'methods and materials' section. |
| Introduction | Introduction | Line 6-11 (page 3) | Too much detail for being this high up in the introduction. Focus on the big picture and then narrow down to your research question. This may be better in the methods section | We acknowledge the concerns raised by the reviewer | We moved the material to the methodology section |
| Introduction | Introduction | Line 28-32 (page 3) | This should be higher in the introduction, and should more directly lead to the conclusion that climate change will influence carbon sequestration through the mechanisms of forest NPP | We acknowledge the concerns raised by the reviewer | We moved the material as suggested by the reviewer |
| Introduction | Introduction | Line 4-5 (page 4) | This line reads as an aside and is discontinuous with the rest of the paragraph. I agree that it is an | We acknowledge the concerns raised by the reviewer | We revised the sentence |

| Main-section of the paper | Sub-section of the paper | Line number (Page No.) | Comment from the referee #1 | Author's response | Author's changes in manuscript |
|---|---|---|---|---|---|
| | | | important concept, so perhaps work to fit this in better as a collateral effect of biomass loss (perhaps mention others and consolidate together). | | |
| Materials and methods | Materials and methods | Line 11 (page 4) | I suggest that you re-order this section to improve the flow of information. Perhaps:
 2.1  Study Sites
 2.2  Teak forests description
 2.3  climate data sources (also, condense description of modelled climate data and climate change into climate data sources)
 2.4  LPJ-GUESS model
 2.5  Model setup | We acknowledge the concerns raised by the reviewer | We re-ordered the information following the suggestion by the reviewer |
| Materials and methods | Materials and methods | Line 12 (page 4) | I think that the site/regional details from the intro should be consolidated and placed here. | We acknowledge the concerns raised by the reviewer | We removed some site/regional details from the introduction. However, we did not bring it to 'study site' section, instead, we took it to 'Description of the Zambezi teak forests' section. Some of the information was deleted from the manuscript |
| Materials and methods | LPJ GUESS model description | Line 2-3 (page 6) | Each one of these scales would need a different optimization, non? | The model can be applied at different scales (local, regional or global). However to get accurate results some parameters in the model have to be changed depending on the scale at which the model is applied. In our study, we applied the model at local scale and some parameters had to be changed from the default global level to the local level. | We added a sentence in the paragraph to clarify that we applied the model at the local scale. |
| Materials and methods | LPJ GUESS model description | Line 5 (page 6) | Are you simulating at multiple scales here? | We simulated at local level | We revised the sentence for clarity |
| Materials and methods | LPJ GUESS model description | Line 11-15 (page 6) | Here, you quickly go into some nitty-gritty details about the model. I am left wondering, why do they | It is important that these nitty-gritty details are shown in this section. They | We added a sentence just before showing |

| Main-section of the paper | Sub-section of the paper | Line number (Page No.) | Comment from the referee #1 | Author's response | Author's changes in manuscript |
|---|---|---|---|---|---|
| | | | need to specify these details? At this point I am trusting that you are not simply providing redundant information that I could learn from the model's documentation - rather, you are setting the reader up for understanding the particular tweaks/calibrations that you performed. However, this may be easier to read if you had a short (1-2 sentence) introduction to this section specifying why you are giving us these details. | show how the parameters that we changed are used in the model. | these details to explain why we showed them. |
| Materials and methods | LPJ GUESS model description | Line 30-33 (page 6), and line 1-4 (page 7) | Again, why do we need all of these details? As a reader I am willing to accept your 'black box' model and am more interested in how your experiments with this model produced interesting and novel results. | We acknowledge the concerns raised by the reviewer | We removed these details from the manuscript and referred the reader to the model's documentation. |
| Materials and methods | Data sources | Line 23-24 (Page 7) | Do you identify these weather stations in an appendix? It may be of interest to other researchers to know the exact networks/stations that you used, particularly if any anomalies are found in the future due to instrument error, or other systemic issues. | We acknowledge the concerns raised by the reviewer | We identified weather stations as supplementary information (Figure S7) by providing a map showing all weather stations in ecological zones I and II that provided local climate data for our study. |
| Materials and methods | Description of the modelled climate data | Line 11 (Page 8) | Not quite sure what this means; perhaps it is an English issue language use. | We acknowledge the concerns raised by the reviewer. | We deleted the sentence from the manuscript. |
| Materials and methods | Description of the modelled climate data | Line 13-14 (Page 8) | This could use some justification. Also be sure to discuss the implications of this resample in the discussion. | We acknowledge the concerns raised by the reviewer. | We indicated in the discussion the implication of using gridded data on the results. |
| Materials and methods | Description of the Zambezi teak forests | Line 21 (Page 8) | Okay, good! You do discuss these forests in detail here. Perhaps the introduction should allude to the variability within these forests, and then clarify here. | Details on the variability within the forests where we focused our study were provided in the 'materials and methods section'. General information on the | We did not make any changes to the manuscript |

| Main-section of the paper | Sub-section of the paper | Line number (Page No.) | Comment from the referee #1 | Author's response | Author's changes in manuscript |
|---|---|---|---|---|---|
| | | | | Zambezi teak forests in the region is sufficient in the introduction | |
| Materials and methods | Description of the Zambezi teak forests | Line 29 (Page 8) | 50% by biomass, # of stems, canopy coverage? | We acknowledge the concerns raised by the reviewer. | We revised the sentence for clarity |
| Materials and methods | Model set-up | Line 9 -11 (Page 9) | I am confused by this sentence. What are you using those variables for? | We used these variables to run the model to determine the historical NPP. Among other purposes, we used NPP values generated using historical climate data from GCMs to determine changes in NPP (Refer to Section 3.3 in the revised manuscript). | No changes were made to the manuscript |
| Materials and methods | Model set-up | Line 24 (Page 9) | Lower case | We acknowledge the concerns raised by the reviewer. | We edited the letter from upper case ('T)' to lower case ('t)'. |
| Materials and methods | Model set-up | Line 29-30 (Page 9) | Why are these different values? | These were the values that were available from the respective local weather stations. | No changes were made to the manuscript |
| Materials and methods | Model set-up | Line 6 (Page 10) | Not sure if this is the correct usage of this word 'Contemporaneously' | We acknowledge the concerns raised by the reviewer. | We used a different word for clarity |
| Results | Results | Not indicated | Fig. 3 could be improved by adding 1:1 lines to help the reader | We acknowledge the concerns raised by the reviewer and we revised figure 3 as suggested. | We added the 1:1 lines to the graphs in figure 3 |
| Results | Results | Not indicated | (Also, 'modelled contemporaneously climate' doesn't make sense to me.) Just because the model has a high $r^2$ doesn't mean that it's particularly good; different metrics like modelling efficiency (Nash & Sutcliffe) may help | We acknowledge the concerns raised by the reviewer and we revised figure 3 as suggested. | We showed the Nash-Sutcliffe efficiency (NSE) values on the graphs. These values provided more information on the performance of the models in addition to $r^2$ and p-values. |
| Results | Projected climatic conditions: RCP 4.5 and RCP 8.5 | Line 11 (Page 10) | I disagree that these should be presented as 'results' - these are values that you derived from existing datasets, and therefore would be more appropriate in the 'methods and materials' section as input data. | We acknowledge the concerns raised by the reviewer. | We moved the information to the 'Materials and methods' section as suggested by the reviewer. |
| Results | The LPJ-GUESS model validation | Line 9 (Page 11) | I'm confused by this statement. What is the statistical test used to | We clearly stated in the same paragraph that we correlated | We included the p-values in the revised |

| Main-section of the paper | Sub-section of the paper | Line number (Page No.) | Comment from the referee #1 | Author's response | Author's changes in manuscript |
|---|---|---|---|---|---|
| | | | determine significance? How is this a validation if the results were not significant? | standardised tree-ring indices with LPJ-GUESS simulated NPP. Relating tree ring indices to model simulated NPP was one of the validation methods we applied in our study and we had to report the results even if the results were not significant. | manuscript to show the results of statistical tests. |
| Results | The LPJ-GUESS model validation | Line 11-15 (Page 11) | I am not an expert with this model, however I don't think it is appropriate to include the results derived from the default settings here. To me, that would be an inappropriate use of the model. Reporting the total error for each site is important, as you do below, thought the 47% error for Namwala is concerning. | We disagree with the reviewer on the suggestion not to report results generated with default parameter values (Please refer to our explanation above).\n\nThe 47% error at Namwala was high, but it reduced when we used local parameter valued compared to using default parameter values. | We did not make any changes to the manuscript |
| Results | LPJ Guess model validation | Line 1-2 (Page 12) | Again, why include default values? It looks like the allometric equations worked well with the local values. | We disagree with the reviewer on the suggestion not to report results generated with default parameter values. Please check our explanation above. | We did not make any changes to the manuscript |
| Results | LPJ Guess model validation | Line 10-12 (Page 12) | Same comment as above re: default values. | We disagree with the reviewer on the suggestion not to report results generated with default parameter values. Please check our explanation above. | We did not make any changes to the manuscript |
| Results | Carbon stocks, LAI and NPP | Line 7-9 (Page 14) | Same comment as above re: default values. | We disagree with the reviewer on the suggestion not to report results generated with default parameter values. Please check our explanation above. | We did not make any changes to the manuscript |
| Results | Climate change effects on NPP | Line 2-3 (Page 15) | These seem to me to be very small changes in NPP. What is the expected error of the model? Are these within the model's margin of error? What are the error bars plotted here? | Please refer to the explanation we provided above on the margin of error for LPJ Guess model.\n\nError bars plotted are the standard deviation (difference between the standard deviation of NPP for the period 1960-1989 and 2070 -2099). | We removed the error bars from the graphs in the revised manuscript. |
| Discussion | The LPJ GUESS model performance | Line 6-11 (Page 15) | Unnecessary in my opinion. | It was necessary for us to emphasis in the discussion how the results were affected after using local parameter values compared to using default parameters. The projected changes in NPP were estimated by using local parameters. This | We did not make any changes to the manuscript |

| Main-section of the paper | Sub-section of the paper | Line number (Page No.) | Comment from the referee #1 | Author's response | Author's changes in manuscript |
|---|---|---|---|---|---|
| | | | | necessitated us to first report how local parameters affected NPP results. | |
| Discussion | The LPJ GUESS model performance | Line 12 (Page 15) | Please report statistical test, and test statistics, for these results. | The sentence clearly indicates that we performed a simple correlation between LPL GUESS simulated NPP and tree ring indices. | We have reported in the revised version the p-values at each site |
| Discussion | The LPJ GUESS model performance | Line 13-17 (Page 15) | These are good points. Could this also be due to the fact that the model produces a mean NPP value (ensemble of all trees), while the individual trees represent the variability present within a forest? e.g. one tree may be restricted in its growth due to competitive pressure, while the overall NPP at the model's resolution includes the more successful trees within its estimates? | The point raised by the reviewer is correct | We included the suggestion of the reviewer in the revised manuscript |
| Discussion | The LPJ GUESS model performance | Line 19 ) Page 15) | Is this the same thing as that site being moisture limited? | Carry-over effects of rainfall on trees does not mean that the site is moisture limited. This means that rainfall of the previous year(s) affect NPP of the current year. This effect can happen at site with either limited moisture or without limited moisture. In our study, we found this effect at site with limited moisture. | We did not make any changes to the manuscript |
| Discussion | LPJ GUESS model performance | Line 3-6 (Page 16) | I think that this is an interesting idea, and you are probably correct, however I do not think that you actually prove this in your study, and therefore this claim is unsupported. | The reviewer is correct that we did not prove the point that increasing the number of tree species in tree-ring analysis would improve the relationship between LPJ-GUESS simulated NPP and tree-ring indices. However, in the sentence that followed, we clearly stated that further studies need to be conducted. It is in this proposed study that our theory can either be proven or rejected. | We did not make any changes to the manuscript |
| Discussion | NPP's climate response | Line 15 (Page 16) | I am unclear if this is a true correlation - did you conduct a statistical test of correlation? If so - include test statistics in results and reiterate here. | In the same sentence, we referred the reader to Supplementary Information Fig. S5 and Supplementary Information Fig. S4. In this supplementary information, the reviewer will find all the statistical tests. We are asking the reviewer to read the supplementary information as well. | We did not make any changes to the manuscript |

| Main-section of the paper | Sub-section of the paper | Line number (Page No.) | Comment from the referee #1 | Author's response | Author's changes in manuscript |
|---|---|---|---|---|---|
| Discussion | NPP's climate response | Line 1 (Page 17) | results | We acknowledge the concerns raised by the reviewer. | We corrected the error from 'result' to 'results' |
| Discussion | NPP's climate response | Line 10 (Page 17) | results | We acknowledge the concerns raised by the reviewer. | We corrected the error from 'result' to 'results' |
| Discussion | NPP's climate response | Line 11 (Page 17) | Again, I dispute the use of the word correlation absent statistical tests. | In the same sentence, we referred the reader to Supplementary Information Fig. S3. In this supplementary information, the reviewer will find all the statistical tests. We are asking the reviewer to read the supplementary information as well. | We did not make any changes to the manuscript |
| Discussion | NPP's climate response | Line 4-7 (Page 18) | This is an interesting point, however I am unsure that this heterogeneity is captured in the model, seeing as the model's taxonomic resolution is at the PFT level | The discussion is supported by information that is captured by the model and also by other studies (or methods) in these forests. Information on the heterogeneity of the trees in the forest was captured through field survey. In the sentence that followed, we provided statistics on the distribution of tree species in these forests. We also indicated the literature where the reviewer can get details on our previous studies. We are therefore requesting the reviewer to read the cited literature for more details on species distribution. Using LPJ GUESS model, our study did not focus on species distribution since we already generated this data through field survey. | We did not make any changes to the manuscript |
| Discussion | NPP's climate response | Line 1 (Page 19) | This statement needs support in the form of a reference or logical argument. | We acknowledge the concerns raised by the reviewer. | We revised the statement |
| Discussion | NPP's climate response | Line 4 (Page 19) | . T (needs a space) | We acknowledge the concerns raised by the reviewer. | We added the space |
| Discussion | NPP's climate response | Line 5-6 (Page 19) | Did charcoal burning influence any of your study sites? If not, that should not impact the accuracy of the model and validation within your specific study areas. This is true, however, if applied to larger regions where charcoal burning is an issue. | Charcoal burning is not a very serious illegal activity in all the three study site and thus did not affect our results. However, during the field survey, we observed two charcoal kilns in Namwala site. Thus, charcoal production can affect model results negatively if the production is done on a large scale, thus, the need to provide for in LPJ-GUESS model. | We did not make any changes to the manuscript |

| Main-section of the paper | Sub-section of the paper | Line number (Page No.) | Comment from the referee #1 | Author's response | Author's changes in manuscript |
|---|---|---|---|---|---|
| Discussion | NPP's climate response | Line 17-18 (Page 19) | Is this a sign that your use of a global model is inappropriate at the local scale? | No. It is just an indication of limited number of researchers using these global models in Africa. This can be seen through publications on the individual researchers who use these models. Most researchers are not based in Africa. This indicates a gap in knowledge on such models between researchers in Africa and those based in other countries. | We did not make any changes to the manuscript |
| Conclusions | Conclusions | Line 24 (Page 19) | suggest the word 'gathered' or 'collected' | We generated soil and tree parameter values from the soil and tree samples that we collected/gathered. The word 'generated' is therefore correct. | We did not make any changes to the manuscript |
| Conclusions | Conclusions | Line 29-31 (Page 19) | It would be good to include % change values here, even if redundant with above. | We acknowledge the concerns raised by the reviewer. | We included the % change in the conclusion as suggested by the reviewer |
| Conclusions | Conclusions | Line 32-33 (Page 19) | I'm not sure that 'rainfall patterns' is appropriate here because I don't think those are really captured in the input data (i.e. if the input data give precip as annual values and coarse spatial resolution, I would consider that climatology and not rainfall patterns, which to me implies finer spatial and temporal scales) | We acknowledge the concerns raised by the reviewer. | We revised the sentence for clarity. |
| Conclusions | Conclusions | Line 1-2 (Page 20) | I suggest that you take this line of reasoning one step further - that $CO_2$ concentrations will be more important in forests that are generally not temperature or precip limited, however precipitation will continue to be the limiting factor in your drier site. | We acknowledge the concerns raised by the reviewer. | We took the reasoning further as suggested by the reviewer |

[revised manuscript text omitted]

---

## Author Response (AR3)

**Responses to anonymous referee # 1**

Thank you for your valuable comments. We incorporated suggestions from the reviewer and we also spent some time and efforts to improve the language in the revised version. Below are our responses to the reviewer and the changes we made to the manuscript. Changes in the revised manuscript are highlighted in yellow.

**General comments from the reviewer**

This is my second review of this manuscript, and I am happy to see that it has been substantially revised in a way that is quite responsive to my previous comments. After reading through the authors' responses, I think that I understand this manuscript much better, and I think that the authors' arguments are well founded and generally supported. I have some suggestions for further revision, though I would like to say that I think this manuscript is much improved.

My main concern at this point is concerning the use of the word correlation. After reading through this again carefully, and considering the authors' responses to my previous review, I think that this term is perhaps being used incorrectly, though it should be fixable and will not likely substantially change the heart of the results. In the manuscript, as well as supplementary figures S1-S5, the authors use the term correlation where I believe they are performing a bivariate linear regression. While they are very similar, their results and interpretation vary. The results shown here have a scatterplot with a best-fit line, as well as associated r2 value and p-value. I presume that these values were generated with a linear model (such as lm() in R). The resulting r2 value represents the amount of variability within the dependent variable (y-axis) that is explained by the independent variable (x-axis). The linear regression is predicated on a causal relationship between the variables (which seems to be true for these comparisons, though it isn't explicitly stated as such). A linear regression will often also report p-values for the slope and intercept, with the null hypothesis being that, separately, the slope and intercept are equal to zero. It appears that the p-value reported herein is for the slope of the line. Therefore, having a p-value which is small for the slope of the line in these bivariate regressions is evidence that the slope is non-zero, though this is not the same thing as having a statistically significant correlation (though one usually goes with the other).

On the other hand, the authors could use a one-tailed Pearson's product-moment correlation test (presuming their data satisfy the test assumptions, otherwise a Spearman's would probably work fine with similar results) to determine if there is a significant correlation with P<0.05 (or their chosen alpha level). It is likely that the results of the correlation test will agree with the slope of the regression line. I

think that your story will stay the same either way, but I would prefer to see the term correlation used to describe a correlation test, or to rephrase the manuscript to describe the linear models appropriately (because they are not technically correlations).

Finally, figures S4 and S5 have a few panels showing P<0; which would be better as P<0.001

5  **Specific comments from the reviewer #1 and authors responses**

| Comment from the referee #1 | Author's response | Author's changes in manuscript |
|---|---|---|
| Refer to general comments from the reviewer under section 1.1 above | We acknowledge the concern raised by the reviewer | We revised the manuscript and the supplementary information. We presented results of spearman's correlations in the revised versions |
| Figures S4 and S5 have a few panels showing P<0; which would be better as P<0.001 (This comment is as presented under section 1.1 above) | We acknowledge the concern raised by the reviewer | We revised Figures S4 and S5 and presented p-values as 'p<0.001' |
| Page 2 Line 15: looks like there are two spaces between: "respectively especially" but maybe that's the justified formatting. | We acknowledge the concern raised by the reviewer | We removed one space |
| Page 2 Line 20: I would remove the comma after 'thus' | We acknowledge the concern raised by the reviewer | We removed the comma after 'thus' |
| Page 3 line 10: I would remove comma after "Reduced rainfall" | We acknowledge the concern raised by the reviewer | We removed comma after "Reduced rainfall" |
| Page 3 line 12: This paragraph begins with a statement about regional variability, however this is followed by statements that seem to describe Africa as a whole. | We acknowledge the concern raised by the reviewer | We revised the first sentence in the paragraph. |
| Page 5 line 8: suggest "80 tree species" or otherwise specify as appropriate. | We acknowledge the concern raised by the reviewer | We included the word 'tree' before '80' in the sentence. |
| Page 13 line 28 – minor suggestion; I think it would be more clear to say that NPP increases at Namwala and Kabompo sites, but decreases at Sesheke | We acknowledge the concern raised by the reviewer | We revised the sentence for clarity as suggested by the reviewer |
| Page 17 Table 3: perhaps I'm missing something, but these changes in NPP % appear to differ from the values listed in the Conclusions (Page 19 section 4.4) | We acknowledge the concern raised by the reviewer | We corrected the figures in Table 3 |

[revised manuscript text omitted]